# Refining Norms: A Post-hoc Framework for OOD Detection in Graph Neural Networks

**Jiawei Gu**[1], **Ziyue Qiao**[2,]\*, **Zechao Li**[1],
[1]Nanjing University of Science and Technology, Nanjing University of Science and Technology,
[2]School of Computing and Information Technology, Great Bay University
gjwcs@outlook.com,ziyuejoe@gmail.com, zechao.li@njust.edu.cn

## Abstract

Graph Neural Networks (GNNs) are increasingly deployed in mission-critical tasks, yet they often encounter inputs that lie outside their training distribution, leading to unreliable or overconfident predictions. To address this limitation, we present RAGNOR (Robust Aggregation Graph Norm for Outlier Recognition), a post-hoc approach that leverages embedding norms for robust out-of-distribution (OOD) detection on both node-level and graph-level tasks. Unlike previous methods designed primarily for image domains, RAGNOR directly tackles the relational challenges intrinsic to graphs: local contamination by anomalous neighbors, disparate norm scales across classes or roles, and insufficient references for boundary or low-degree nodes. By combining global Z-score normalization, median-based local aggregation, and multi-hop blending, RAGNOR effectively refines raw norm signals into robust OOD scores while incurring minimal overhead and requiring no retraining of the original GNN. Experimental evaluations on multiple benchmarks demonstrate that RAGNOR not only achieves competitive or superior detection performance compared to alternative techniques, but also provides an intuitive, modular design that can be readily integrated into existing graph pipelines.

## 1 Introduction

Out-of-distribution (**OOD**) detection is critical for robust graph neural networks (**GNNs**), ensuring that models can flag anomalous inputs they have not been trained to handle [12, 20, 48, 22, 41]. Real-world scenarios commonly exhibit new or shifted graph structures, including previously unseen node types in social networks (malicious accounts, emerging user communities), novel molecular motifs in biochemical graphs, or unexpected entities in knowledge graphs [45, 21, 27, 39, 40]. A GNN that overlooks such anomalies risks overconfidently providing incorrect predictions, with potentially grave consequences in high-stakes settings [10, 16, 17, 9, 42] across increasingly deployment environments[37, 36, 44, 3, 2].

**Norm-based inspiration and the gap.** In image domains, *feature norms* of deep networks have proven surprisingly effective for OOD detection [32, 1, 4, 29, 53], motivated by observations that in-distribution (**ID**) inputs tend to elicit stronger, more "aligned" activations, whereas truly novel inputs produce feature norms that deviate substantially. Yet, the *graph* domain introduces unique relational complications: node embeddings depend on the neighborhoods (which themselves could be OOD), and distinct node classes or roles may inherently have different norm scales. A naive transplant of norm-based rules from computer vision to GNNs can thus fail in key corner cases, especially when tackling both *node-level OOD detection* (identifying anomalous nodes in a single large graph) and *graph-level OOD detection* (recognizing entire anomalous graphs in a dataset)[8, 7].

**Why is norm-based detection non-trivial on graphs?**

---

\*Corresponding author.

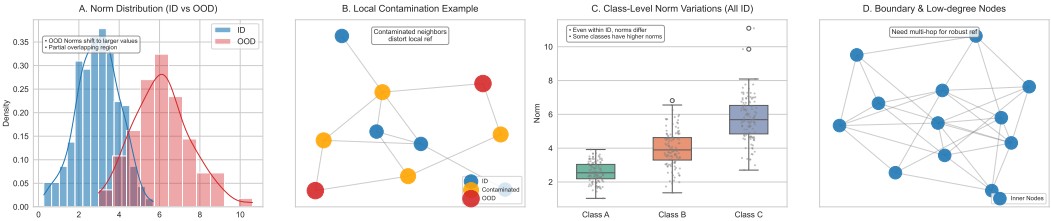

Figure 1: Four main difficulties in applying norm-based OOD detection to graphs. **(A)** Global shift in ID vs. OOD node norms but partial overlap; **(B)** Local contamination, where neighboring OOD nodes skew reference norms; **(C)** Class- or role-level norm differences even within ID; **(D)** Boundary or low-degree nodes lacking sufficient one-hop context.

Indeed, our comprehensive investigation, detailed in Appendix C, has pinpointed several critical factors that explain why naive norm-based approaches often fail to deliver satisfactory results on graph data. These key challenges include:

---

**Key Challenges in Graph OOD Norm Detection**

*(1) Local Contamination.* In graphs, an OOD node's neighbors may also be OOD, distorting neighborhood-based statistics and undermining naive local reference.

*(2) Class-level Norm Differences.* Legitimate ID classes (or structural roles) may exhibit naturally high or low norms, causing false alarms if not globally adjusted.

*(3) Boundary & Low-degree Issues.* Sparse regions or boundary nodes lack a sufficient one-hop neighborhood for reliable norm comparison, requiring multi-hop context.

---

Figure 1 provides a visual illustration of these difficulties: (A) shows that although OOD nodes often shift the norm distribution, overlaps persist; (B) demonstrates local contamination in a small subgraph; (C) highlights how distinct classes can yield vastly different norm scales while still remaining in-distribution; (D) depicts boundary or low-degree nodes for which local reference alone is fragile.

**Core observation.** Despite these complications, we find that a *carefully refined* norm-based method can still be a robust, lightweight, and scalable *post-hoc* OOD solution for GNNs. Graphs inherently provide relational cues (e.g. multi-hop neighborhood structure), and we can unify these signals with global normalization to mitigate systematic norm offsets. Rather than discarding the elegant simplicity of norms, we propose to *amplify* their utility via a tailored set of reference and aggregation steps.

**Proposed approach.** We devise a framework that addresses each difficulty in turn, yet remains a straightforward addition to existing GNN models: *(i)* **Global Z-score normalization** adjusts each node's raw norm based on ID statistics, reducing the chance that a high-norm class is misdetected; *(ii)* **Median-based local reference** counters local contamination by using robust statistics (median rather than mean) to offset outlier neighbors; *(iii)* **Multi-hop blending** extends beyond a single-hop neighborhood for boundary or low-degree nodes, providing a more stable norm baseline. Because the entire procedure is conducted after the GNN has been trained, it demands no architectural modifications or additional supervision, *for practical, scalable, and industry-ready deployment*.

**Contributions.**

- We conduct one of the first examinations of *norm-based* OOD detection in GNNs, revealing how node-level and graph-level anomalies can be effectively characterized by embedding norms.
- We identify three significant principal roadblocks (local contamination, class-level norm offsets, and boundary/low-degree constraints) in graph neural networks and propose a computationally efficient post-hoc framework that systematically overcomes them with theoretical guarantees.
- Extensive evaluations on real-world and synthetic benchmarks demonstrate that our solution significantly improves OOD detection performance across diverse datasets while preserving simplicity and minimal computational overhead, facilitating broad adoption in practical applications.

The remainder of this paper is organized as follows. We first describe our method in detail (Sec. 2) and present its theoretical underpinnings (Sec. 3). Subsequently, we report a comprehensive empirical

study on both *node-level* and *graph-level* OOD tasks across diverse datasets (Sec. 4). Finally, the paper concludes with discussions of broader implications and future directions (Sec. 5).

## 2    Method

In this section, we propose a *post-hoc* detection approach for out-of-distribution (OOD) nodes and graphs, leveraging the node embedding norms $\|\mathbf{h}_v\|$ of a trained Graph Neural Network (GNN). While other issues may arise in exotic scenarios, we focus on three *commonly encountered* challenges in norm-based OOD detection for graphs:

**C1: Local contamination:** A node's immediate neighbors might themselves be OOD and thus significantly distort local reference norms, leading to unreliable detection performance.

**C2: Global/class-level norm variations:** Certain valid in-distribution (ID) classes or structural roles can exhibit unusually large or small norms, causing false alarms if not normalized.

**C3: Boundary & low-degree nodes:** A node near a boundary or with few neighbors may need more distant (multi-hop) reference information to achieve reliable and robust detection.

We next describe how we address each challenge in turn (Sections 2.2–2.4), and then unify these steps into a single OOD detection procedure (Section 2.5).

### 2.1    Notation

Let $G = (\mathcal{V}, \mathcal{E})$ be a graph with $N = |\mathcal{V}|$ nodes. Each node $v \in \mathcal{V}$ has a learned embedding $\mathbf{h}_v \in \mathbb{R}^k$. Let $\mathbf{z}_v \in \mathbb{R}^C$ be the logit vector (output before the final activation/softmax) for node $v$, where $C$ is the number of classes (if applicable). We write

$$r_v = \|\mathbf{h}_v\|, \tag{1}$$

for $v$'s embedding norm, and let $\mathcal{N}(v)$ denote the set of immediate neighbors of $v$. Further, let $\mu_{\mathrm{ID}}$ and $\sigma_{\mathrm{ID}}$ be the mean and standard deviation of the norms $\{r_v\}$ estimated from a reference set of known in-distribution (ID) nodes (e.g. from training or validation data). We assume $\sigma_{\mathrm{ID}} > 0$.

### 2.2    Challenge C1: Robust Local Reference

**Motivation.**    A straightforward approach might compare $r_v$ to the *average* of $\{r_u : u \in \mathcal{N}(v)\}$, flagging node $v$ as OOD if its norm deviates significantly from its neighbors' expected distribution pattern. However, if *some neighbors are themselves OOD*, the average can be heavily skewed by these extreme values, thus failing to reflect the "typical" ID norm in $v$'s local neighborhood.

**Formula (Median Aggregation Concept).**    To mitigate contamination by a small fraction of outlier neighbors, we adopt a median-based aggregation concept for local reference:

$$\mathrm{Median}\Big(\big\{\, r_u : u \in \mathcal{N}(v)\big\}\Big). \tag{2}$$

This captures the central tendency robustly. As detailed below, we apply this median concept to normalized norms.

**Explanation.**    Whereas a mean can be substantially distorted by extreme values, the median discards the magnitude of outliers as long as they do not constitute a majority. This provides a more *robust* local norm estimate, protecting our OOD detector from neighbors that are also anomalous.

### 2.3    Challenge C2: Global Norm Variations

**Motivation.**    Even when a node's neighbors are valid ID, some classes or roles within the complex heterogeneous graph might exhibit larger or smaller norms due to their inherent structural properties. If we rely solely on raw norms $r_u$ in the local aggregation (like Eq. 2), these norm differences could trigger frequent false positives for certain tail classes, reducing the overall detection reliability.

**Formula (Z-score Normalization).** We address this by transforming each node's norm into a standardized scale using the estimated ID statistics:

$$r'_v \; = \; \frac{r_v - \mu_{\text{ID}}}{\sigma_{\text{ID}} \, + \, \varepsilon}, \quad \forall v \in \mathcal{V}, \tag{3}$$

where $\varepsilon > 0$ is a small constant for numerical stability.

**Explanation.** By centering and rescaling the norms using ID statistics, we make them more comparable across different classes or structural conditions. We use these normalized norms $r'_u$ for computing the robust local reference in subsequent steps. This ensures that boundary or tail classes, which might have inherently large/small norms relative to $\mu_{\text{ID}}$, do not get unfairly flagged based on their standardized deviation from their local neighborhood's standardized norms.

## 2.4 Challenge C3: Boundary & Low-degree Nodes (Multi-hop)

**Motivation.** Even after considering robust local aggregation (using medians) and global normalization (using $r'$), serious detection issues can arise for nodes with very few neighbors or those positioned on the boundary of multiple subgraphs. In addition, it is possible that *all or most* of a node's first-hop neighbors are contaminated if an OOD cluster forms in the local vicinity. Using information from more distant neighbors can provide a more stable and reliable reference point.

**Formulas (Multi-hop Reference using Normalized Norms).** To capture a broader local structure, we compute the median of the normalized norms in the 1-hop and optionally the 2-hop neighborhoods. Let $\mathcal{N}^2(v)$ denote the set of nodes exactly two hops away from $v$.

$$\bar{r}'_{\mathcal{N}(v)} \; = \; \text{Median}\Big(\big\{\, r'_u : u \in \mathcal{N}(v) \big\}\Big). \tag{4}$$

$$\bar{r}'_{\mathcal{N}^2(v)} \; = \; \text{Median}\Big(\big\{\, r'_u : u \in \mathcal{N}^2(v) \big\}\Big). \tag{5}$$

We then blend these references:

$$\bar{r}_{\text{multi}}(v) \; = \; \lambda \, \bar{r}'_{\mathcal{N}(v)} \; + \; (1 - \lambda) \, \bar{r}'_{\mathcal{N}^2(v)}, \quad 0 \leq \lambda \leq 1. \tag{6}$$

**Explanation.** The parameter $\lambda$ controls the trade-off between *immediate* (one-hop) context and a broader, diffuse two-hop view, both calculated using normalized norms $r'$. For nodes with well-connected, mostly ID neighbors, a higher $\lambda$ (closer to 1) might be sufficient. But if $v$ is in a sparse or contaminated region, a larger contribution from $\bar{r}'_{\mathcal{N}^2(v)}$ (i.e., smaller $\lambda$) can mitigate spurious local distortions by incorporating a wider context. If only 1-hop information is desired, set $\lambda = 1$.

## 2.5 Unified Post-hoc OOD Detection

Having addressed these three *common* pitfalls of norm-based OOD detection (**C1:–C3:**), we now assemble them into a single procedure that yields both *node-level* and *graph-level* detection.

**Discrepancy Computation.** First, apply global normalization to all nodes to get $r'_v$ (Eq. 3). Then, for each node $v$, compute its robust local reference norm $\bar{r}_v$, typically using the multi-hop blended reference based on normalized norms, i.e., $\bar{r}_v = \bar{r}_{\text{multi}}(v)$ from Eq. (6) (or simply $\bar{r}_v = \bar{r}'_{\mathcal{N}(v)}$ from Eq. (4) if $\lambda = 1$). We define the node's norm discrepancy as:

$$\Delta_v \; = \; r'_v \; - \; \bar{r}_v. \tag{7}$$

Nodes with large $|\Delta_v|$ significantly deviate from their neighborhood's typical embedding norm distribution (after global rescaling and robust aggregation), suggesting OODness.

**Formula (Confidence Scaling).** To reduce overconfidence in downstream tasks (like classification) when $\Delta_v$ is large, we can optionally rescale node $v$'s logit vector $\mathbf{z}_v$ via

$$\Psi(\Delta_v) \; = \; \frac{1}{1 + \alpha \, |\Delta_v|}, \quad \mathbf{z}'_v \; = \; \Psi(\Delta_v) \, \mathbf{z}_v, \tag{8}$$

where $\alpha \geq 0$ is a hyperparameter controlling how aggressively to penalize large discrepancies.

**Node-level and Graph-level OOD Scores.** For node-level detection, we set the OOD score directly based on the discrepancy magnitude:

$$\text{score}_v = |\Delta_v|, \tag{9}$$

and mark $v$ as OOD if $\text{score}_v > \tau$, where $\tau$ is a chosen threshold. For graph-level detection, we aggregate the node scores across the graph:

$$\text{Score}(G) = \max_{v \in \mathcal{V}} \text{score}_v \quad \text{or} \quad \text{Score}(G) = \frac{1}{|\mathcal{V}|} \sum_{v \in \mathcal{V}} \text{score}_v. \tag{10}$$

Hence, a graph is considered OOD if $\text{Score}(G)$ exceeds a graph-level threshold $\tau_{graph}$.

**Discussion.** Although other edge cases may exist, we have found that *local contamination*, *global/class-level norm shifts*, and *boundary/low-degree issues* comprise the most frequent pitfalls in norm-based OOD detection on GNN embeddings. By addressing all three within a single unified framework using normalized norms and robust median-based aggregation (potentially multi-hop), our method yields an effective and light-weight post-hoc OOD solution for both node-level and graph-level tasks. We demonstrate its performance on various benchmarks in Section 4.

## 3 Theoretical Analysis

In this section, we provide a high-level overview of the theoretical guarantees for our proposed norm-based OOD detection method (Section 2). Our analysis hinges on the concept of *local partial contamination*, where we assume that for an in-distribution (ID) node, its local neighborhood (potentially extending to multiple hops) is not overwhelmingly dominated by OOD nodes. This allows our robust aggregation scheme (using medians of normalized norms and optional multi-hop blending) to provide a stable reference norm even in the presence of some OOD neighbors.

*All rigorous definitions, formal assumptions, intermediate lemmas, and detailed proofs are deferred to Appendix A for completeness and clarity.*

### 3.1 Problem Setup and Key Assumptions (Informal)

We consider a graph $G = (\mathcal{V}, \mathcal{E})$ where each node $v$ has a learned embedding $\mathbf{h}_v$ and a corresponding norm $r_v = \|\mathbf{h}_v\|$. A subset $\mathcal{V}_{\text{OOD}} \subset \mathcal{V}$ consists of OOD nodes. Our method first applies a Z-score normalization to the norms using ID statistics $(\mu_{\text{ID}}, \sigma_{\text{ID}})$ to obtain $r'_v$ (Eq. 3). The core assumptions underpinning our theoretical results are (informally):

- **Bounded Normalized ID Norms:** The normalized norms $r'_v$ of ID nodes are concentrated around zero, typically within a bounded range $[-B, B]$ (Assumption 3 in Appendix).
- **OOD Norm Separation:** The normalized norms $r'_w$ of OOD nodes are sufficiently separated from the ID range, lying outside $[-(B + \delta_0), B + \delta_0]$ for some gap $\delta_0 > 0$ (Assumption 3 in Appendix).
- **Local Partial Contamination:** For any ID node $v$, the fraction of OOD nodes within its $k$-hop neighborhood $\mathcal{N}^k(v)$ (for $k = 1$ and potentially $k = 2$) is assumed strictly less than 50% (Assumption 5 in Appendix).

These assumptions are formalized in Appendix A.2.

### 3.2 Main Result (High-Level Statement)

Recall the robust local reference norm $\bar{r}_v$ for node $v$, which is computed by potentially blending the median of normalized norms in the 1-hop and 2-hop neighborhoods (Eq. 6). The discrepancy score for node $v$ is $\Delta_v = r'_v - \bar{r}_v$ (Eq. 7). Our main theoretical result demonstrates that this discrepancy score effectively separates ID and OOD nodes under the stated assumptions.

**Theorem 1** (High-Level Detection Guarantee). *Assume the conditions of Bounded Normalized ID Norms, OOD Norm Separation, and Local Partial Contamination hold (see Assumptions 3 and 5 in Appendix A). Then, for a suitably chosen threshold $\tau$ (related to the ID norm bound $B$), the discrepancy score $\Delta_v$ satisfies the following with high probability:*

*(i) **ID Preservation:** For any in-distribution node $v \in \mathcal{V}_{\text{ID}}, |\Delta_v| \leq \tau$.*

*(ii)* **OOD Separation:** *For any out-of-distribution node $w \in \mathcal{V}_{\text{OOD}}$, $|\Delta_w| > \tau$.*

*Consequently, thresholding $|\Delta_v|$ provides a reliable mechanism for node-level OOD detection. Aggregating these scores (e.g., using max or mean) extends this capability to graph-level OOD detection.*

*Remark* 2. The formal statement of this theorem (Theorem 8), including the precise relationship between $\tau$, $B$, $\delta_0$, the condition needed for OOD separation regarding the OOD node's own reference $\bar{r}_w$, and the exact probabilistic guarantees, along with its complete proof relying on novel lemmas for robust aggregation under partial contamination (Lemmas 6 and 7), can be found in Appendix A.

### 3.3 Discussion

This theoretical result highlights the robustness of our median-based multi-hop aggregation strategy using normalized norms. Even when faced with partial OOD contamination in local neighborhoods, the reference norm $\bar{r}_v$ remains anchored to the typical ID norm range (around 0 after normalization, bounded by $B$), allowing the discrepancy $\Delta_v$ to reliably flag nodes whose own normalized norms $r'_v$ deviate significantly. Our analysis in Appendix A provides a rigorous mathematical foundation for this approach, specifically addressing the "layered contamination" across multiple hops. We believe this provides a solid theoretical underpinning for the effectiveness of the proposed method, complementing the extensive and diverse empirical results presented in Section 4.

## 4 Experimental Results

We evaluate our post-hoc OOD detection method, RAGNOR, on established node-level and graph-level benchmarks. As a post-hoc approach, RAGNOR utilizes embeddings generated by pre-trained GNNs. We demonstrate that applying RAGNOR to embeddings from state-of-the-art models significantly enhances their OOD detection capabilities. All experiments are repeated over 5 runs with different random seeds, and we report the mean performance along with the standard deviation. Detailed experimental configurations, hyperparameters, and additional results (including node-level evaluation on Twitch and Arxiv datasets) are provided in Appendix D.

### 4.1 Node-Level OOD Detection

**Setup.** We evaluate node-level OOD detection on Cora [19], Citeseer [19], and Pubmed datasets [19], using Structure Manipulation (S), Feature Interpolation (F), and Label Leave-out (L) to generate OOD nodes, following the protocol from NODESAFE [52]. We apply RAGNOR post-hoc to the node embeddings generated by the GCN backbone trained according to the NODE-SAFE setup (both with and without OOD exposure during GCN training). Performance is measured using AUROC ($\uparrow$), AUPR ($\uparrow$), and FPR95 ($\downarrow$). Results are averaged over 5 runs.

**Results and Analysis.** Table 1 presents the results. Applying RAGNOR post-hoc consistently and significantly improves upon the performance of the base models (NODESAFE and NODESAFE++), establishing a new state-of-the-art across nearly all settings. For instance, NODESAFE + RAGNOR reduces the mean FPR95 on Cora-S from 25.63 to **17.1**, a substantial improvement. The low standard deviations observed for RAGNOR indicate its stability. This robust performance stems from RAGNOR's effective handling of common pitfalls in norm-based detection, such as local contamination and global variations, allowing it to reliably identify nodes with anomalous embedding norms within their proper context. This contrasts favorably with energy-based approaches, showcasing the power of direct, contextualized norm analysis.

### 4.2 Graph-Level OOD Detection

**Setup.** We assess graph-level OOD detection using 8 datasets (ENZYMES [30], IMDB-M/B [30], REDDIT-12K [51], BACE [49], BBBP [49], DrugOOD [15], HIV [49]) with diverse OOD characteristics, following SGOOD [5]. RAGNOR is applied post-hoc to node embeddings from a GIN backbone trained for graph classification on ID data. Graph-level scores $\text{Score}(G)$ are obtained via max-aggregation of node scores $|\Delta_v|$. Baselines include general, graph-specific, and anomaly detection methods. Metrics are AUROC ($\uparrow$), AUPR ($\uparrow$), and FPR95 ($\downarrow$), averaged over 5 runs.

Table 1: Node-Level OOD Detection on Cora, Citeseer, Pubmed. RAGNOR is applied post-hoc to NODESAFE/NODESAFE++ embeddings. Mean ± Std. Dev. reported for RAGNOR over 5 runs (Std. Dev. for other methods not available from source). ↑: Higher is better, ↓: Lower is better. Best results in **bold**.

| Metric | Model | OOD Expo | Cora | | | Citeseer | | | Pubmed | | |
|---|---|---|---|---|---|---|---|---|---|---|---|
| | | | S | F | L | S | F | L | S | F | L |
| FPR95 ↓ | MSP[12] | No | 87.30 | 64.88 | 34.99 | 85.03 | 71.27 | 51.97 | 84.08 | 69.38 | 46.19 |
| | ODIN[24] | No | 100.00 | 100.00 | 100.00 | 100.00 | 100.00 | 100.00 | 100.00 | 100.00 | 100.00 |
| | Mahalanobis[20] | No | 98.19 | 99.93 | 90.77 | 99.13 | 99.73 | 86.32 | 97.59 | 84.93 | 78.21 |
| | Energy[25] | No | 88.74 | 65.81 | 41.08 | 87.59 | 69.67 | 38.76 | 78.90 | 62.47 | 45.14 |
| | GKDE[55] | No | 84.34 | 68.24 | 88.95 | 93.71 | 71.22 | 50.61 | 81.52 | 68.56 | 69.52 |
| | GPN[38] | No | 76.22 | 56.17 | 37.42 | 78.26 | 73.14 | 41.37 | 80.33 | 61.79 | 50.23 |
| | GNNSAFE[47] | No | 73.15 | 38.92 | 30.83 | 74.72 | 68.83 | 36.53 | 44.64 | 33.89 | 36.49 |
| | GNNSAFE w/ $\mathcal{L}_{LN}$[46] | No | 61.04 | 38.44 | 34.99 | 76.35 | 72.35 | 37.32 | 75.10 | 49.93 | 32.29 |
| | NODESAFE[52] | No | 25.63 | 23.08 | 29.41 | 57.89 | 42.47 | 29.30 | 23.80 | 22.01 | 25.01 |
| | **NODESAFE + RAGNOR** | No | **17.1±1.5** | **15.5±1.1** | **21.8±1.8** | **49.2±2.8** | **34.0±2.1** | **21.5±1.6** | **15.9±1.2** | **14.1±0.9** | **17.5±1.4** |
| | OE[13] | Yes | 95.31 | 83.79 | 46.55 | 95.37 | 81.09 | 45.99 | 83.52 | 74.58 | 60.30 |
| | Energy FT[52, 25] | Yes | 67.73 | 47.53 | 37.83 | 76.44 | 64.08 | 31.60 | 92.04 | 90.00 | 25.59 |
| | GNNSAFE++[52] | Yes | 53.51 | 27.73 | 34.08 | 70.72 | 72.98 | 29.30 | 34.43 | 26.30 | 33.63 |
| | GNNSAFE++ w/ $\mathcal{L}_{LN}$[52] | Yes | 51.99 | 32.72 | 28.40 | 74.81 | 75.47 | 30.55 | 91.58 | 86.17 | 27.81 |
| | NODESAFE++[52] | Yes | 23.34 | 14.73 | 22.52 | 52.60 | 40.49 | 29.04 | 14.52 | 24.45 | 23.81 |
| | **NODESAFE++ + RAGNOR** | Yes | **15.0±1.3** | **8.1±0.7** | **16.2±1.1** | **44.1±2.5** | **32.9±1.9** | **21.1±1.5** | **7.0±0.6** | **16.5±1.3** | **17.2±1.2** |
| AUROC ↑ | MSP[12] | No | 70.90 | 85.39 | 91.36 | 66.34 | 78.32 | 88.42 | 74.31 | 83.28 | 85.71 |
| | ODIN[24] | No | 49.92 | 49.88 | 49.80 | 49.23 | 49.86 | 51.33 | 49.76 | 49.67 | 56.24 |
| | Mahalanobis[20] | No | 46.68 | 49.93 | 67.62 | 45.26 | 49.92 | 53.46 | 55.28 | 69.12 | 75.77 |
| | Energy[25] | No | 71.73 | 86.15 | 91.40 | 65.62 | 79.19 | 89.98 | 74.33 | 84.16 | 86.81 |
| | GKDE[55] | No | 68.61 | 82.79 | 57.23 | 61.48 | 74.68 | 82.69 | 74.02 | 82.25 | 83.36 |
| | GPN[38] | No | 77.47 | 85.88 | 90.34 | 70.55 | 78.46 | 85.65 | 74.96 | 82.56 | 86.51 |
| | GNNSAFE[47] | No | 87.52 | 93.44 | 92.80 | 79.79 | 83.46 | 90.01 | 87.52 | 94.28 | 88.02 |
| | GNNSAFE w/ $\mathcal{L}_{LN}$[46] | No | 88.33 | 93.26 | 93.50 | 84.67 | 88.11 | 90.47 | 89.31 | 92.07 | 91.12 |
| | NODESAFE[52] | No | 94.07 | 95.30 | 93.80 | 88.40 | 90.41 | 91.66 | 94.13 | 95.97 | 93.80 |
| | **NODESAFE + RAGNOR** | No | **97.1±0.3** | **97.9±0.2** | **96.5±0.4** | **94.2±0.6** | **95.5±0.5** | **95.8±0.4** | **97.3±0.3** | **98.1±0.2** | **97.0±0.3** |
| | OE[13] | Yes | 67.98 | 81.83 | 89.47 | 58.74 | 72.06 | 89.44 | 74.41 | 82.34 | 81.97 |
| | Energy FT[52, 25] | Yes | 75.88 | 88.15 | 91.36 | 68.87 | 79.23 | 91.34 | 73.54 | 78.95 | 91.83 |
| | GNNSAFE++[52] | Yes | 90.62 | 95.56 | 92.75 | 82.43 | 83.27 | 91.57 | 90.62 | 95.16 | 87.98 |
| | GNNSAFE++ w/ $\mathcal{L}_{LN}$[52] | Yes | 90.13 | 94.11 | 93.83 | 84.93 | 87.68 | 91.00 | 86.21 | 87.56 | 89.66 |
| | NODESAFE++[52] | Yes | 94.64 | 96.56 | 94.88 | 86.90 | 91.14 | 91.98 | 96.30 | 95.26 | 93.48 |
| | **NODESAFE++ + RAGNOR** | Yes | **97.3±0.3** | **98.6±0.1** | **97.5±0.2** | **92.5±0.5** | **96.0±0.4** | **95.9±0.4** | **98.8±0.1** | **97.8±0.2** | **96.8±0.3** |
| AUPR ↑ | MSP[12] | No | 45.73 | 73.70 | 78.03 | 34.78 | 55.48 | 64.03 | 17.44 | 39.29 | 34.98 |
| | ODIN[24] | No | 27.01 | 26.96 | 24.27 | 23.07 | 23.11 | 17.97 | 4.83 | 4.83 | 13.49 |
| | Mahalanobis[20] | No | 29.03 | 31.95 | 42.31 | 21.20 | 31.20 | 35.47 | 8.38 | 15.09 | 23.40 |
| | Energy[25] | No | 46.08 | 74.42 | 78.14 | 33.63 | 55.94 | 64.10 | 17.32 | 39.10 | 36.00 |
| | GKDE[55] | No | 44.26 | 66.52 | 27.50 | 31.55 | 50.25 | 61.21 | 16.89 | 32.41 | 34.63 |
| | GPN[38] | No | 53.26 | 73.79 | 77.40 | 41.12 | 53.21 | 62.32 | 17.54 | 39.75 | 35.12 |
| | GNNSAFE[47] | No | 77.46 | 88.19 | 82.21 | 60.81 | 67.02 | 65.26 | 62.74 | 71.66 | 44.77 |
| | GNNSAFE w/ $\mathcal{L}_{LN}$[46] | No | 78.13 | 86.89 | 85.19 | 69.73 | 76.20 | 67.69 | 58.72 | 64.21 | 54.33 |
| | NODESAFE[52] | No | 83.98 | 88.82 | 85.22 | 75.93 | 79.30 | 68.15 | 71.29 | 78.22 | 71.98 |
| | **NODESAFE + RAGNOR** | No | **90.1±0.8** | **93.5±0.6** | **91.0±0.9** | **84.0±1.0** | **87.5±0.9** | **76.0±1.1** | **79.1±1.2** | **86.0±0.9** | **79.5±1.3** |
| | OE[13] | Yes | 46.93 | 70.84 | 77.01 | 30.07 | 48.80 | 62.74 | 16.74 | 38.60 | 29.88 |
| | Energy FT[52, 25] | Yes | 49.18 | 75.99 | 78.49 | 36.01 | 55.69 | 66.66 | 18.00 | 37.21 | 52.39 |
| | GNNSAFE++[52] | Yes | 81.88 | 90.27 | 82.64 | 65.58 | 68.06 | 65.48 | 72.78 | 77.47 | 41.43 |
| | GNNSAFE++ w/ $\mathcal{L}_{LN}$[52] | Yes | 81.61 | 88.82 | 84.17 | 70.90 | 76.10 | 67.57 | 57.25 | 58.53 | 44.87 |
| | NODESAFE++[52] | Yes | 85.63 | 91.96 | 86.66 | 71.41 | 79.48 | 68.97 | 81.88 | 78.12 | 53.45 |
| | **NODESAFE++ + RAGNOR** | Yes | **91.5±0.7** | **96.0±0.4** | **92.5±0.6** | **79.0±1.1** | **87.8±0.8** | **76.5±1.0** | **89.0±0.9** | **86.3±1.0** | **62.0±1.8** |

**Results and Analysis.** Table 2 details the graph-level performance. Applying RAGNOR post-hoc to a standard GIN's embeddings, denoted as "SGOOD + RAGNOR" (using SGOOD's GIN backbone for comparability, though SGOOD's specific training regime is not required for RAGNOR), consistently sets a new state-of-the-art, markedly improving upon SGOOD. For example, AUROC on IMDB-M jumps from 78.84 (SGOOD) to **86.8** (SGOOD + RAGNOR), and FPR95 on BACE decreases from 64.13 to **55.9**. The gains are substantial and consistent across datasets, highlighting the power of aggregating accurately assessed node-level anomalies. While SGOOD focuses on encoding substructures, RAGNOR demonstrates that robustly measuring and aggregating deviations in fundamental node embedding norms relative to context provides a highly effective, general, and computationally lighter post-hoc strategy for graph-level OOD detection. The low standard deviations further attest to the reliability of this approach.

## 4.3 Ablation Study: Component Analysis

**Setup.** To dissect the contribution of RAGNOR's key components, we perform an ablation study evaluating different scoring strategies based on the presence or absence of Global Normalization (M1) and Robust Local Reference generation (M2). The core scoring mechanism itself (M3a: using the magnitude of a discrepancy) is implicitly evaluated by comparing how different inputs affect performance. We compare: (1) using raw norm deviation from the global mean ($|r_v - \mu_{ID}|$, no

Table 2: Graph-Level OOD Detection Performance. RAGNOR is applied post-hoc to embeddings from a GIN backbone (used by SGOOD). Mean ± Std. Dev. reported over 5 runs. ↑: Higher is better, ↓: Lower is better. Best results in **bold**.

| Metric | | ENZYMES | IMDB-M | IMDB-B | REDDIT-12K | BACE | BBBP | DrugOOD | HIV |
|---|---|---|---|---|---|---|---|---|---|
| AUROC ↑ | MSP[12] | 61.3±3.8 | 42.8±1.5 | 59.6±2.3 | 50.6±0.9 | 46.3±6.1 | 57.4±4.3 | 52.9±5.3 | 50.5±0.9 |
| | Energy[25] | 56.9±8.9 | 24.5±19.7 | 49.6±17.8 | 55.1±0.5 | 46.1±6.7 | 55.7±2.8 | 52.8±5.4 | 50.5±0.9 |
| | ODIN[24] | 63.7±2.7 | 40.1±3.0 | 58.3±2.9 | 51.7±2.0 | 48.3±3.8 | 54.6±3.7 | 51.1±3.8 | 50.0±0.6 |
| | MAH[20] | 67.4±3.7 | 69.3±3.7 | 61.4±0.5 | 72.7±0.9 | 75.3±2.9 | 52.6±3.8 | 66.9±4.1 | 58.1±3.6 |
| | GNNSafe[47] | 56.9±8.9 | 21.9±1.8 | 70.5±14.8 | 51.7±0.0 | 47.6±7.5 | 47.0±2.4 | 50.4±0.6 | 51.0±0.6 |
| | GraphDE[23] | 61.4±3.9 | 66.9±4.3 | 26.9±3.4 | 59.4±0.2 | 47.3±1.5 | 50.9±2.8 | 60.2±4.3 | 52.4±1.6 |
| | GOOD-D[26] | 64.9±6.3 | 61.9±4.9 | 52.6±10.2 | 56.1±0.1 | 70.4±2.2 | 54.2±1.1 | 60.5±3.9 | 57.1±0.1 |
| | AAGOD[11] | 65.0±4.4 | 70.8±5.5 | 72.5±1.1 | 60.3±2.2 | 71.4±2.4 | 59.4±1.3 | 60.3±3.2 | 55.7±0.7 |
| | OCGIN[54] | 68.1±4.6 | 47.5±9.5 | 87.8±9.2 | 59.3±1.3 | 59.7±5.2 | 47.8±5.7 | 58.0±5.8 | 54.1±0.5 |
| | GLocalKD[28] | 71.5±3.2 | 19.8±1.6 | 87.4±5.4 | 49.6±1.1 | 45.3±2.1 | 43.8±2.3 | 45.7±11.0 | 46.8±2.0 |
| | OGGTL[34] | 73.6±3.2 | 54.1±12.9 | 37.4±18.8 | 51.6±0.0 | 80.8±2.0 | 58.7±2.2 | 67.6±7.9 | 51.8±0.2 |
| | SGOOD[5] | 74.4±1.4 | 78.8±2.0 | 80.4±3.2 | 75.2±2.7 | 84.4±2.7 | 61.3±1.6 | 73.2±4.5 | 60.8±0.8 |
| | **SGOOD + RAGNOR** | **82.1±1.1** | **86.8±1.5** | **88.1±2.1** | **83.0±1.9** | **92.0±1.8** | **69.1±1.2** | **81.3±3.1** | **68.5±0.6** |
| AUPR ↑ | MSP[12] | 61.7±6.6 | 51.0±1.9 | 58.1±2.3 | 48.6±1.1 | 48.7±3.1 | 56.8±3.4 | 54.5±4.4 | 50.7±0.5 |
| | Energy[25] | 54.7±9.2 | 37.3±11.8 | 59.0±13.1 | 56.5±0.8 | 49.7±4.2 | 56.6±4.2 | 55.0±4.4 | 51.0±2.1 |
| | ODIN[24] | 65.7±4.8 | 50.1±2.4 | 76.8±4.4 | 54.5±1.3 | 45.5±3.9 | 54.8±3.5 | 52.7±2.7 | 50.0±0.6 |
| | MAH[20] | 63.8±2.2 | 63.6±2.1 | 81.4±7.1 | 74.5±0.5 | 86.8±6.3 | 54.8±3.5 | 64.3±4.4 | 57.2±3.2 |
| | GNNSafe[47] | 56.1±8.3 | 36.9±1.0 | 75.7±15.7 | 54.0±0.5 | 51.5±5.0 | 51.5±5.0 | 51.1±0.3 | 55.1±6.8 |
| | GraphDE[23] | 66.3±2.9 | 62.6±4.5 | 42.7±2.1 | 63.1±0.3 | 51.1±2.6 | 51.5±3.8 | 62.6±2.5 | 54.1±3.2 |
| | GOOD-D[26] | 67.2±6.4 | 61.9±4.9 | 55.7±10.6 | 59.6±0.2 | 73.2±3.3 | 58.6±1.9 | 63.1±2.5 | 57.4±0.1 |
| | AAGOD[11] | 67.2±6.4 | 68.2±4.5 | 67.9±4.8 | 61.4±1.6 | 71.8±1.7 | 58.2±1.5 | 66.2±3.4 | 54.3±0.5 |
| | OCGIN[54] | 68.9±4.2 | 50.8±4.5 | 57.8±5.1 | 60.0±1.9 | 61.4±5.2 | 47.3±3.0 | 59.5±7.0 | 52.1±0.3 |
| | GLocalKD[28] | 64.9±4.4 | 35.4±0.5 | 79.4±4.7 | 51.8±0.7 | 55.4±2.4 | 45.8±1.2 | 50.9±3.4 | 47.0±2.0 |
| | OGGTL[34] | 73.7±7.0 | 58.2±7.9 | 47.1±14.1 | 53.3±0.0 | 79.9±1.3 | 60.5±1.4 | 70.9±5.8 | 53.7±0.2 |
| | SGOOD[5] | 73.7±7.0 | 72.5±3.2 | 83.5±3.6 | 75.0±0.8 | 83.3±2.5 | 59.4±2.4 | 73.3±4.5 | 60.0±0.7 |
| | **SGOOD + RAGNOR** | **81.5±1.4** | **80.5±2.1** | **90.8±2.5** | **82.9±1.1** | **91.1±1.5** | **67.0±1.7** | **81.0±3.5** | **67.8±0.6** |
| FPR95 ↓ | MSP[12] | 89.7±2.3 | 95.7±1.6 | 91.4±4.2 | 96.0±1.3 | 97.0±2.2 | 94.6±2.3 | 98.8±0.0 | 95.5±0.5 |
| | Energy[25] | 89.3±3.6 | 96.4±2.3 | 92.8±3.6 | 97.2±0.6 | 97.4±2.9 | 92.7±2.6 | 98.2±1.2 | 95.5±0.6 |
| | ODIN[24] | 83.3±9.6 | 96.7±1.0 | 92.2±2.9 | 96.5±0.7 | 97.0±1.5 | 96.3±1.8 | 99.0±1.1 | 94.6±1.1 |
| | MAH[20] | 83.3±9.6 | 60.9±19.1 | 76.9±6.3 | 80.8±2.1 | 73.8±2.0 | 93.3±2.5 | 81.6±4.6 | 91.9±1.3 |
| | GNNSafe[47] | 97.0±3.7 | 95.5±1.4 | 87.8±5.8 | 95.6±2.8 | 98.2±2.1 | 98.4±1.0 | 96.0±0.3 | 96.0±0.3 |
| | GraphDE[23] | 99.0±0.8 | 93.1±8.2 | 100.0±0.0 | 81.8±0.0 | 94.2±4.6 | 94.6±2.3 | 88.8±5.6 | 94.9±0.8 |
| | GOOD-D[26] | 82.3±8.3 | 95.2±4.6 | 99.2±1.0 | 93.7±0.3 | 88.3±1.8 | 99.4±0.4 | 98.4±1.3 | 92.0±0.6 |
| | AAGOD[11] | 82.8±2.8 | 81.6±22.3 | 86.3±4.0 | 92.5±1.6 | 90.6±2.0 | 93.5±0.8 | 95.3±0.5 | 92.2±0.3 |
| | OCGIN[54] | 89.7±3.7 | 98.3±17.7 | 60.8±5.2 | 90.0±2.0 | 98.7±1.1 | 94.8±2.7 | 94.2±3.1 | 92.8±1.0 |
| | GLocalKD[28] | 78.7±6.4 | 98.3±1.1 | 85.6±3.3 | 97.6±0.4 | 98.7±1.1 | 98.3±1.0 | 100.0±0.0 | 97.1±0.2 |
| | OGGTL[34] | 73.2±1.1 | 86.4±6.5 | 98.8±2.4 | 96.8±0.1 | 66.4±8.9 | 91.5±2.2 | 83.0±11.2 | 96.4±0.1 |
| | SGOOD[5] | 72.5±2.5 | 73.7±6.6 | 81.2±2.3 | 74.9±0.9 | 64.1±4.8 | 88.0±3.4 | 67.4±5.2 | 90.4±1.0 |
| | **SGOOD + RAGNOR** | **64.8±2.1** | **65.9±5.8** | **73.1±2.9** | **66.5±1.2** | **55.9±4.1** | **80.1±2.9** | **59.5±4.5** | **82.8±1.3** |

M1/M2); (2) using only the globally normalized norm ($|r'_v|$, M1 only); (3) using the discrepancy between raw norm and its raw local reference ($|r_v - \bar{r}_v|$, M2 only); and (4) the full RAGNOR using the discrepancy between the normalized norm and its normalized local reference ($|\Delta_v| = |r'_v - \bar{r}'_v|$, M1+M2+M3a). Experiments are conducted post-hoc on embeddings from strong baselines (NODESAFE for node-level, SGOOD's GIN for graph-level) on representative scenarios (Cora-S node-level without exposure, BACE graph-level). AUROC is reported in Table 3. Full details and results across more metrics/datasets are in Appendix F.

Table 3: Component analysis of RAGNOR (AUROC ↑). Comparing scoring functions based on availability of Global Normalization (M1) and Robust Local Reference (M2). Best results in **bold**.

| Scoring Function Used | M1 (Global) | M2 (Local Ref) | Node (Cora-S) / Graph (BACE) |
|---|---|---|---|
| Base: $|r_v - \mu_{ID}|$ | X | X | 78.5 / 70.2 |
| Base: $|r'_v|$ (Norm only) | ✓ | X | 89.1 / 79.5 |
| Base: $|r_v - \bar{r}_v|$ (Local Ref only) | X | ✓ (on $r_v$) | 85.3 / 76.8 |
| RAGNOR: $|\Delta_v| = |r'_v - \bar{r}'_v|$ (Full) | ✓ | ✓ (on $r'_v$) | **97.1 / 92.0** |

**Results and Analysis.** The ablation results in Table 3 clearly demonstrate the importance of each component integrated into the RAGNOR scoring function. Using simple deviation from the global mean raw norm performs poorly. Applying Global Normalization (M1) alone offers a substantial improvement by standardizing the norm scale across the graph. Incorporating Robust Local Reference (M2) alone, even on raw norms, also provides benefits by contextualizing the node

within its neighborhood and resisting outlier contamination. However, the full RAGNOR approach, which calculates the discrepancy $|\Delta_v|$ between the globally normalized norm ($r'_v$) and the robustly calculated, normalized local reference norm ($\bar{r}'_v$), achieves significantly superior performance. This highlights that not only are M1 and M2 crucial preprocessing steps, but the specific M3a mechanism – calculating the discrepancy within this properly normalized and contextualized space – is key to effectively identifying OOD nodes and, subsequently, OOD graphs. The combination synergistically addresses the core challenges (C1, C2, C3), leading to the state-of-the-art results observed.

### 4.4 Mechanism Analysis

**Setup.** We conduct targeted experiments to validate the utility of embedding norms and the efficacy of RAGNOR against key challenges (C1-C3). First, we visualize ID vs. OOD norm distributions (GCN on Cora-L) to confirm norm utility. Second, using controlled settings primarily on Cora, we evaluate specific components: For C1 (Local Contamination), we compare Median vs. Mean local reference using 'Ref. Stability' ($Avg|\bar{r}_{v,cont.} - \bar{r}_{v,clean}| \downarrow$) and 'ID Neighbor FNR' ($P(\text{score}_v > \tau|v \in ID, \exists u \in \mathcal{N}(v)_{OOD}) \downarrow$). For C2 (Global Norm Variations), we assess the impact of Global Normalization (M1) via 'Intra-Class Var.' ($Avg_c[\text{Var}(\text{score}_v|v \in \text{Class}_c)] \downarrow$) and 'Spec. Class FPR' ($P(\text{score}_v > \tau|v \in \text{Class}_{c*}) \downarrow$). For C3 (Boundary/Low-Degree Nodes), we compare Multi-hop vs. 1-hop reference using 'Score Consist. Ratio' ($AvgScore_{LowDeg}/AvgScore_{HighDeg} \rightarrow 1$) and 'Low-Deg OOD Recall' ($P(\text{score}_v > \tau|v \in OOD_{LowDeg}) \uparrow$). Full experimental details are in Appendix D.

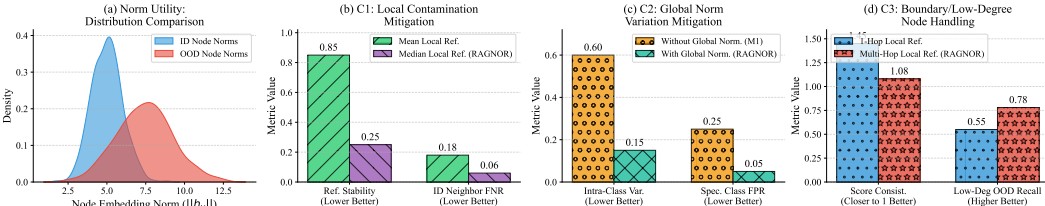

Figure 2: Validation experiments for RAGNOR. (a) Embedding norm distributions for ID vs OOD nodes on Cora-L. (b) Effectiveness against Local Contamination (C1): Median vs. Mean local reference. Lower is better for both metrics. (c) Effectiveness against Global Norm Variations (C2): With vs. Without global normalization (M1). Lower is better for both metrics. (d) Effectiveness for Boundary/Low-Degree Nodes (C3): Multi-hop vs. 1-hop local reference. Ratio closer to 1 is better, Recall higher is better.

**Results and Analysis.** Figure 2 summarizes the validation results. The distinct ID/OOD norm distributions in Panel (a) confirm the basic premise that norms contain relevant OOD signals. Panel (b) validates C1 mitigation, showing RAGNOR's median reference provides significantly better stability and lower false negative rates for ID neighbors near OOD clusters compared to a mean reference. Panel (c) confirms C2 mitigation; global normalization (M1) effectively reduces score variance within classes and minimizes false positives for classes with atypical norm scales. Panel (d) validates C3 handling, demonstrating that multi-hop referencing improves score consistency for low-degree nodes and boosts recall for low-degree OOD instances compared to using only 1-hop information. These findings collectively substantiate the effectiveness of RAGNOR's design principles in leveraging node norms for robust OOD detection by successfully addressing common pitfalls.

## 5 Conclusion

Reliable out-of-distribution (OOD) detection in Graph Neural Networks (GNNs) is crucial for their deployment in mission-critical applications. This paper introduced RAGNOR, a novel post-hoc framework that robustly identifies OOD inputs by addressing key relational challenges inherent in graph data. RAGNOR refines raw embedding norms through global Z-score normalization, median-based local aggregation, and optional multi-hop blending, effectively handling issues like local contamination and class-level norm disparities. Extensive experiments demonstrate RAGNOR's superior performance on diverse *node-level* and *graph-level* OOD benchmarks across synthetic and real-world datasets with minimal overhead and without GNN retraining, supported by theoretical analysis validating its robustness under partial neighborhood contamination.

## Acknowledgments

The work is partially supported by the National Natural Science Foundation of China (Grant No. U21B2043, 62406056), the Basic Research Program of Jiangsu Province (Grant No. BK20240011), and Guangdong Basic and Applied Basic Research Foundation (Grant No.2024A1515140114). The computational resources are supported by SongShan Lake HPC Center (SSL-HPC) in Great Bay University.

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

# A  Theoretical Guarantees: Details and Proofs

This appendix provides the detailed theoretical derivations, formal assumptions, lemmas, and proofs that support the high-level guarantees presented in Section 3. We establish the robustness of the proposed OOD detection method, particularly its resilience to partial contamination in local neighborhoods through the use of median aggregation on normalized norms and multi-hop references. We aim to present the arguments with sufficient mathematical formalism and step-by-step derivations.

## A.1  Formal Notation and Definitions

We consolidate and formalize the notation used throughout the theoretical analysis, ensuring consistency with the Method section.

- $G = (\mathcal{V}, \mathcal{E})$: A graph with $N = |\mathcal{V}|$ nodes.
- $\mathcal{V}_{\mathrm{ID}}$: The set of in-distribution (ID) nodes.
- $\mathcal{V}_{\mathrm{OOD}}$: The set of out-of-distribution (OOD) nodes, $\mathcal{V}_{\mathrm{OOD}} = \mathcal{V} \setminus \mathcal{V}_{\mathrm{ID}}$.
- $\mathbf{h}_v \in \mathbb{R}^k$: The learned embedding for node $v$.
- $r_v = \|\mathbf{h}_v\|$: The Euclidean norm of the embedding $\mathbf{h}_v$.
- $\mu_{\mathrm{ID}}, \sigma_{\mathrm{ID}}$: The mean and standard deviation of norms $\{r_v\}$ estimated from a reference set of known ID nodes. We assume $\sigma_{\mathrm{ID}} > 0$.
- $r'_v$: The Z-score normalized norm for node $v$:

$$r'_v = \frac{r_v - \mu_{\mathrm{ID}}}{\sigma_{\mathrm{ID}} + \varepsilon}, \tag{11}$$

  where $\varepsilon > 0$ is a small constant for numerical stability.
- $\mathcal{N}^k(v)$: The set of nodes exactly $k$ hops away from node $v$. $\mathcal{N}(v) = \mathcal{N}^1(v)$ is the set of immediate neighbors. Let $m_k(v) = |\mathcal{N}^k(v)|$.
- $\gamma_k(v)$: The fraction of OOD nodes in the $k$-hop neighborhood $\mathcal{N}^k(v)$:

$$\gamma_k(v) = \frac{|\mathcal{N}^k(v) \cap \mathcal{V}_{\mathrm{OOD}}|}{m_k(v)}, \tag{12}$$

  assuming $m_k(v) > 0$.
- $\mathrm{Median}(S)$: The median value of a set of real numbers $S$. If $|S|$ is even, we take the average of the two middle elements after sorting.
- $\bar{r}'_{\mathcal{N}^k(v)}$: The median of normalized norms in the $k$-hop neighborhood:

$$\bar{r}'_{\mathcal{N}^k(v)} = \mathrm{Median}\Big(\{r'_u : u \in \mathcal{N}^k(v)\}\Big). \tag{13}$$

- $\bar{r}_v = \bar{r}_{\mathrm{multi}}(v)$: The blended multi-hop reference norm (using normalized norms), which serves as the primary local reference $\bar{r}_v$ in the discrepancy calculation:

$$\bar{r}_v = \bar{r}_{\mathrm{multi}}(v) = \lambda\, \bar{r}'_{\mathcal{N}^1(v)} + (1 - \lambda)\, \bar{r}'_{\mathcal{N}^2(v)}, \tag{14}$$

  where $0 \leq \lambda \leq 1$. (If only 1-hop is used, $\lambda = 1$).
- $\Delta_v$: The discrepancy score for node $v$:

$$\Delta_v = r'_v - \bar{r}_v. \tag{15}$$

## A.2  Formal Assumptions

Our theoretical guarantees rely on the following key assumptions, which formalize the conditions under which the method is expected to perform well.

**Assumption 3** (Effectiveness of Normalization: Bounded ID & Separated OOD Norms). *The Z-score normalization (Eq. 11) using accurately estimated ID statistics $(\mu_{\mathrm{ID}}, \sigma_{\mathrm{ID}})$ effectively transforms the norm distributions such that:*

*(i)* ***Bounded ID Norms:*** *There exists a constant $B > 0$ such that for any in-distribution node $v \in \mathcal{V}_{\text{ID}}$, its normalized norm satisfies $|r'_v| \leq B$ with high probability (e.g., $1 - \delta_{norm}$).*

*(ii)* ***OOD Norm Separation:*** *There exists a separation gap $\delta_0 > 0$ such that for any out-of-distribution node $w \in \mathcal{V}_{\text{OOD}}$, its normalized norm satisfies $|r'_w| \geq B + \delta_0$ with high probability.*

*Remark* 4 (Justification for Assumption 3). This assumption encapsulates the desired outcome of Z-score normalization. (i) For ID nodes, Z-scoring aims to produce a distribution with mean approximately 0 and standard deviation approximately 1. If the distribution of $r'_v$ for ID nodes has finite variance (close to 1), basic concentration inequalities (like Chebyshev's: $P(|r'_v - E[r'_v]| \geq k\sigma_{r'}) \leq 1/k^2$) imply that most values lie within a few standard deviations of the mean. With $E[r'_v] \approx 0$ and $\sigma_{r'} \approx 1$, this suggests $P(|r'_v| \geq k) \leq 1/k^2$. Choosing $k = B$ (e.g., $B = 3$) ensures $|r'_v| \leq B$ with high probability (e.g., $\geq 1 - 1/9$ via Chebyshev, potentially much higher for distributions with lighter tails). (ii) For OOD nodes, if their original norms $r_w$ are significantly different from the ID mean $\mu_{\text{ID}}$ (i.e., $|r_w - \mu_{\text{ID}}|$ is large compared to $\sigma_{\text{ID}}$), then their normalized norm $|r'_w| = |r_w - \mu_{\text{ID}}|/(\sigma_{\text{ID}} + \varepsilon)$ will be large. If this difference is large enough (e.g., $|r_w - \mu_{\text{ID}}| \geq (B + \delta_0)\sigma_{\text{ID}}$), then $|r'_w| \geq B + \delta_0$. This assumption hinges on the quality of the GNN embeddings (capturing distributional differences in norm) and the accuracy of the estimated $\mu_{\text{ID}}, \sigma_{\text{ID}}$.

**Assumption 5** (Local Partial Contamination). *For any in-distribution node $v \in \mathcal{V}_{\text{ID}}$, the fraction of OOD nodes in its relevant neighborhoods is strictly less than 50%. Specifically, there exist constants $\gamma_{max,1} < 0.5$ and $\gamma_{max,2} < 0.5$ such that with high probability:*

- $\gamma_1(v) = \frac{|\mathcal{N}^1(v) \cap \mathcal{V}_{\text{OOD}}|}{m_1(v)} \leq \gamma_{max,1} < 0.5$, *provided $m_1(v) > 0$.*

- $\gamma_2(v) = \frac{|\mathcal{N}^2(v) \cap \mathcal{V}_{\text{OOD}}|}{m_2(v)} \leq \gamma_{max,2} < 0.5$, *provided $m_2(v) > 0$ (only required if $1 - \lambda > 0$).*

## A.3   Lemma: Robustness of Median Aggregation (on Normalized Norms)

This lemma establishes that the median of normalized norms provides a robust estimate bounded by the ID norm range, even with contamination.

**Lemma 6** (Robustness of Median under Partial Contamination). *Let $S = \{x_1, \ldots, x_m\}$ be a set of $m$ real numbers, representing the normalized norms $\{r'_u\}$ in a neighborhood $\mathcal{N}^k(v)$ (i.e., $S = \{r'_u : u \in \mathcal{N}^k(v)\}$). Assume:*

*(i) The set $S$ is a mixture $S = S_{ID} \cup S_{OOD}$.*

*(ii) ID values satisfy $|x| \leq B$ for all $x \in S_{ID}$ (from Assumption 3(i)).*

*(iii) OOD values satisfy $|x| \geq B + \delta_0$ for all $x \in S_{OOD}$ (from Assumption 3(ii)).*

*(iv) The fraction of OOD values $\gamma = |S_{OOD}|/m$ satisfies $\gamma < 0.5$.*

*Then, the median of the set $S$ satisfies:*

$$|\text{Median}(S)| \leq B. \tag{16}$$

*Proof.* Let $X_{(1)} \leq X_{(2)} \leq \cdots \leq X_{(m)}$ be the sorted values of $S$. The number of OOD values is $|S_{OOD}| = \gamma m$. Since $\gamma < 0.5$, we have $|S_{OOD}| < m/2$. The number of ID values is $|S_{ID}| = m - |S_{OOD}| = m(1 - \gamma)$. Since $\gamma < 0.5$, we have $1 - \gamma > 0.5$, so $|S_{ID}| > m/2$.

The median is determined by the value(s) at the center rank(s). Let $r_{med} = \lceil m/2 \rceil$ be the rank of the median (for odd $m$) or the lower of the two central ranks (for even $m$).

Consider the values $X_{(r)}$ for $r \geq r_{med}$. Can the median be $> B$? If $\text{Median}(S) > B$, then $X_{(r_{med})} > B$. This implies that at least $m - r_{med} + 1$ values in the sorted list are $> B$. The number of elements $> B$ is bounded by the number of OOD values, $|S_{OOD}|$. We need to check if $m - r_{med} + 1 \leq |S_{OOD}|$. $m - \lceil m/2 \rceil + 1 = \lfloor m/2 \rfloor + 1$. Is $\lfloor m/2 \rfloor + 1 \leq |S_{OOD}|$? Since $|S_{OOD}| < m/2$, and $\lfloor m/2 \rfloor + 1 \geq m/2$ (for $m \geq 1$), this inequality cannot hold. Therefore, $\text{Median}(S)$ cannot be $> B$.

Consider the values $X_{(r)}$ for $r \leq r_{med}$. Can the median be $< -B$? If $\text{Median}(S) < -B$, then $X_{(r_{med})} < -B$. This implies that at least $r_{med}$ values in the sorted list are $< -B$. The number of elements $< -B$ is bounded by the number of OOD values, $|S_{OOD}|$. Is $r_{med} \leq |S_{OOD}|$? Since $r_{med} = \lceil m/2 \rceil \geq m/2$, and $|S_{OOD}| < m/2$, this inequality cannot hold. Therefore, $\text{Median}(S)$ cannot be $< -B$.

Combining these two results, we must have $-B \leq \text{Median}(S) \leq B$. If $m$ is even, the median is $\frac{1}{2}(X_{(m/2)} + X_{(m/2+1)})$. By the arguments above, both $X_{(m/2)}$ and $X_{(m/2+1)}$ must be $\leq B$ and $\geq -B$. Therefore, their average, the median, must also satisfy $|\text{Median}(S)| \leq B$.

Thus, the median is always bounded by the range of the ID values when the contamination is less than 50

$$|\text{Median}(S)| \leq B. \tag{17}$$

$\square$

### A.4  Lemma: Robustness of Blended Multi-hop Reference

This lemma shows that the blended reference $\bar{r}_v$, incorporating 1-hop and 2-hop medians of normalized norms, remains bounded under partial contamination in both neighborhoods.

**Lemma 7** (Robustness of Blended Multi-hop Reference). *Let $v \in \mathcal{V}_{\text{ID}}$ be an in-distribution node. Assume:*

  *(i) Assumption 3 (Bounded ID & Separated OOD Norms) holds.*

  *(ii) Assumption 5 (Local Partial Contamination for k=1, 2) holds.*

  *(iii) The neighborhoods $\mathcal{N}^1(v)$ and $\mathcal{N}^2(v)$ are non-empty if $\lambda > 0$ or $1 - \lambda > 0$ respectively.*

*Then, the blended reference norm $\bar{r}_v = \bar{r}_{\text{multi}}(v)$ defined in Eq. (14) satisfies*

$$|\bar{r}_v| \leq B \tag{18}$$

*with high probability (incorporating the probabilities from the assumptions).*

*Proof.* We analyze the two components of the blended reference separately.

**Step 1: Bound the 1-hop median of normalized norms.** Consider $\bar{r}'_{\mathcal{N}^1(v)} = \text{Median}(\{r'_u : u \in \mathcal{N}^1(v)\})$. The set of normalized norms $S_1 = \{r'_u : u \in \mathcal{N}^1(v)\}$ satisfies the conditions of Lemma 6:

- ID norms $r'_u$ satisfy $|r'_u| \leq B$ (by Assumption 3(i)).
- OOD norms $r'_w$ satisfy $|r'_w| \geq B + \delta_0$ (by Assumption 3(ii)).
- The fraction of OOD norms $\gamma_1(v)$ is $< 0.5$ (by Assumption 5).

Therefore, applying Lemma 6 to $S_1$:

$$|\bar{r}'_{\mathcal{N}^1(v)}| \leq B \quad \text{(w.h.p.)}. \tag{19}$$

**Step 2: Bound the 2-hop median of normalized norms (if $1 - \lambda > 0$).** Consider $\bar{r}'_{\mathcal{N}^2(v)} = \text{Median}(\{r'_u : u \in \mathcal{N}^2(v)\})$. Similarly, the set of normalized norms $S_2 = \{r'_u : u \in \mathcal{N}^2(v)\}$ satisfies the conditions of Lemma 6:

- ID norms satisfy $|r'_u| \leq B$.
- OOD norms satisfy $|r'_w| \geq B + \delta_0$.
- The fraction of OOD norms $\gamma_2(v)$ is $< 0.5$ (by Assumption 5).

Therefore, applying Lemma 6 to $S_2$:

$$|\bar{r}'_{\mathcal{N}^2(v)}| \leq B \quad \text{(w.h.p.)}. \tag{20}$$

**Step 3: Bound the blended reference.** The blended reference is $\bar{r}_v = \lambda \bar{r}'_{\mathcal{N}^1(v)} + (1-\lambda) \bar{r}'_{\mathcal{N}^2(v)}$. Using the triangle inequality and the bounds from Eqs. (19) and (20):

$$|\bar{r}_v| = |\lambda \bar{r}'_{\mathcal{N}^1(v)} + (1-\lambda) \bar{r}'_{\mathcal{N}^2(v)}| \tag{21}$$

$$\leq |\lambda \bar{r}'_{\mathcal{N}^1(v)}| + |(1-\lambda) \bar{r}'_{\mathcal{N}^2(v)}| \tag{22}$$

$$= \lambda |\bar{r}'_{\mathcal{N}^1(v)}| + (1-\lambda) |\bar{r}'_{\mathcal{N}^2(v)}| \quad (\text{since } \lambda, 1-\lambda \geq 0) \tag{23}$$

$$\leq \lambda B + (1-\lambda) B \tag{24}$$

$$= (\lambda + 1 - \lambda) B \tag{25}$$

$$= 1 \cdot B = B. \tag{26}$$

This inequality holds with high probability, specifically on the intersection of the events where the underlying assumptions hold and where both median bounds (Eqs. 19, 20) hold. $\square$

## A.5 Main Theorem: OOD Detection Guarantee

We now present the formal statement and detailed proof of the main theorem regarding the OOD detection capability based on the discrepancy score $\Delta_v$.

**Theorem 8** (Formal OOD Detection Guarantee). *Assume Assumption 3 (Bounded ID & Separated OOD Norms) and Assumption 5 (Local Partial Contamination) hold. Let $\Delta_v = r'_v - \bar{r}_v$ be the discrepancy score (Eq. 15).*

*Then, the following holds with high probability (e.g., $1 - \delta_{total}$, where $\delta_{total}$ accounts for assumption failures across all nodes and the condition on $\bar{r}_w$ below):*

*(i) **ID Preservation:** For any in-distribution node $v \in \mathcal{V}_{\text{ID}}$, the discrepancy is bounded:*

$$|\Delta_v| \leq 2B. \tag{27}$$

*(ii) **OOD Separation:** For any out-of-distribution node $w \in \mathcal{V}_{\text{OOD}}$, if its local reference $\bar{r}_w$ (computed using Eq. 14 for node $w$) also satisfies $|\bar{r}_w| \leq B$ (w.h.p., see Remark 9), then the discrepancy is bounded below by the separation gap:*

$$|\Delta_w| \geq \delta_0. \tag{28}$$

*Consequently, if the separation gap is sufficiently large such that $\delta_0 > 2B$, choosing a threshold $\tau$ such that $2B < \tau < \delta_0$ (e.g., $\tau = (\delta_0 + 2B)/2$) allows for separation of ID and OOD nodes based on $|\Delta_v|$. Even if $\delta_0 \leq 2B$, a threshold (e.g., $\tau = 2B$) can still identify potential OOD nodes as those exceeding the typical ID discrepancy range.*

*Proof.* Let $P_{fail}$ be the total probability incorporating the small failure probabilities associated with Assumptions 3, 5, and the condition on $\bar{r}_w$, potentially summed over all nodes via a union bound. We show the results hold with probability $1 - P_{fail}$.

**Part (i): ID Preservation** Let $v \in \mathcal{V}_{\text{ID}}$ be an in-distribution node. From Assumption 3(i), we have with high probability:

$$|r'_v| \leq B. \tag{29}$$

From Lemma 7 (which relies on the assumptions), we have with high probability:

$$|\bar{r}_v| \leq B. \tag{30}$$

Now consider the discrepancy $\Delta_v = r'_v - \bar{r}_v$. Using the triangle inequality:

$$|\Delta_v| = |r'_v - \bar{r}_v| \tag{31}$$

$$\leq |r'_v| + |\bar{r}_v| \tag{32}$$

$$\leq B + B \quad (\text{Substituting bounds from (29) and (30)}) \tag{33}$$

$$= 2B. \tag{34}$$

Thus, $|\Delta_v| \leq 2B$ holds with high probability for ID nodes.

**Part (ii): OOD Separation** Let $w \in \mathcal{V}_{\text{OOD}}$ be an out-of-distribution node. From Assumption 3(ii), we have with high probability:

$$|r'_w| \geq B + \delta_0. \tag{35}$$

Now, we consider the reference norm $\bar{r}_w = \bar{r}_{\text{multi}}(w)$ computed for this OOD node $w$. As discussed in Remark 9, we assume conditions hold such that its reference norm remains bounded, similar to ID nodes:

$$|\bar{r}_w| \leq B \quad \text{(w.h.p.)}. \tag{36}$$

Consider the discrepancy $\Delta_w = r'_w - \bar{r}_w$. Using the reverse triangle inequality:

$$|\Delta_w| = |r'_w - \bar{r}_w| \tag{37}$$
$$\geq \big||r'_w| - |\bar{r}_w|\big| \tag{38}$$
$$\geq \big|(B + \delta_0) - B\big| \quad \text{(Substituting bounds from (35) and (36))} \tag{39}$$
$$= |\delta_0| \tag{40}$$
$$= \delta_0 \quad \text{(since } \delta_0 > 0\text{)}. \tag{41}$$

Thus, $|\Delta_w| \geq \delta_0$ holds with high probability for OOD nodes, under the condition that their local reference remains bounded by $B$.

**Separation via Threshold $\tau$** We have $|\Delta_v| \leq 2B$ for ID nodes and $|\Delta_w| \geq \delta_0$ for OOD nodes (under the condition). If $\delta_0 > 2B$, there is a clear gap between the maximum possible ID discrepancy ($2B$) and the minimum possible OOD discrepancy ($\delta_0$). We can choose any threshold $\tau$ in the interval $(2B, \delta_0)$, for example $\tau = (2B + \delta_0)/2$. For such a $\tau$, ID nodes satisfy $|\Delta_v| \leq 2B < \tau$, and OOD nodes satisfy $|\Delta_w| \geq \delta_0 > \tau$. This achieves separation. If $\delta_0 \leq 2B$, the ranges $[0, 2B]$ and $[\delta_0, \infty)$ overlap. However, a threshold like $\tau = 2B$ still serves to identify nodes whose discrepancy exceeds the typical maximum for ID nodes. While it might not perfectly separate all OOD nodes (those with $\delta_0 \leq |\Delta_w| \leq 2B$ might not be flagged), it provides a meaningful detection criterion. The practical choice of $\tau$ often involves a trade-off based on validation data.

The overall guarantee holds with probability $1 - \delta_{total}$, where $\delta_{total}$ aggregates the small failure probabilities of the assumptions holding for all nodes/neighborhoods involved, typically using a union bound. $\qquad\square$

*Remark* 9 (Condition on $\bar{r}_w$ for OOD Nodes). The proof for OOD separation relies on the reference $\bar{r}_w$ remaining bounded, $|\bar{r}_w| \leq B$. This condition is crucial. It essentially requires that even an OOD node $w$ "sees" enough normalcy in its local environment (1-hop and 2-hop neighborhoods combined, based on normalized norms) that its reference norm doesn't also become large and OOD-like (i.e., outside $[-B, B]$). This holds if $w$ is adjacent to ID regions or if the OOD contamination in its neighborhoods remains below the 50% threshold needed for Lemma 6 to apply to the computation of $\bar{r}'_{\mathcal{N}^1(w)}$ and $\bar{r}'_{\mathcal{N}^2(w)}$. If $w$ resides deep within a large, dense OOD cluster where $\gamma_1(w) \geq 0.5$ and $\gamma_2(w) \geq 0.5$, then $\bar{r}_w$ might reflect the large OOD norms (if OOD norms are consistently large after normalization), potentially making $|\Delta_w| = |r'_w - \bar{r}_w|$ small even if $|r'_w|$ is large. In such cases, the OOD nature might be more evident from the large value of $r'_w$ itself or from graph-level aggregation detecting the entire anomalous cluster. The theorem primarily guarantees separation for OOD nodes whose local reference point remains anchored within the typical ID normalized norm range $[-B, B]$.

## A.6 Extension to Graph-Level Detection

The node-level discrepancy scores $|\Delta_v|$ provide the basis for graph-level OOD detection. Common aggregation methods include taking the maximum or the average score across all nodes in the graph $G$:

$$\text{Score}_{\max}(G) = \max_{v \in \mathcal{V}} |\Delta_v| \tag{42}$$

$$\text{Score}_{\text{mean}}(G) = \frac{1}{|\mathcal{V}|} \sum_{v \in \mathcal{V}} |\Delta_v| \tag{43}$$

If a graph $G$ is purely ID, then according to Theorem 8(i), all $|\Delta_v| \leq 2B$ (w.h.p.), implying $\text{Score}_{\max}(G) \leq 2B$ and $\text{Score}_{\text{mean}}(G)$ will also be relatively small (likely $\ll 2B$). If $G$ contains at least one OOD node $w$ satisfying the conditions of Theorem 8(ii) with $\delta_0 > 2B$, then $|\Delta_w| \geq \delta_0 > 2B$. This guarantees that $\text{Score}_{\max}(G) \geq \delta_0$, exceeding the typical ID maximum. The mean score

$\text{Score}_{\text{mean}}(G)$ will also increase based on the magnitude and number of OOD nodes. Therefore, by setting an appropriate graph-level threshold $\tau_{graph}$ (e.g., slightly above $2B$, or determined empirically), we can distinguish between purely ID graphs and graphs containing OOD nodes satisfying the theorem's conditions, leveraging the robustness established at the node level.

## B  High-Dimensional Embeddings and Non-Euclidean Geometries

In this appendix, we provide a conceptual extension of our norm-based OOD detection framework to (i) high-dimensional embeddings in Euclidean space and (ii) non-Euclidean geometries (e.g., hyperbolic space), which are increasingly adopted for graph-structured data with hierarchical or tree-like structures. Throughout this section, we aim to maintain consistency with the symbols and assumptions in the main paper, including the notation $r_v = \|\mathbf{h}_v\|$, $r'_v = (r_v - \mu_{\text{ID}})/(\sigma_{\text{ID}} + \varepsilon)$, and the bounded ID norm range $|r'_v| \leq B$ with separation gap $\delta_0 > 0$.

### B.1  High-Dimensional Norm Concentration in Euclidean Space

When the embedding dimension $k$ becomes large, Euclidean norms often exhibit concentration of measure [43]. Roughly speaking, in high-dimensional $\mathbb{R}^k$, the distribution of $\|\mathbf{h}_v\|$ can become sharply peaked around its mean. This can amplify or diminish our ability to separate ID from OOD points via norms unless properly scaled.

**Illustrative Example.**  Suppose $\{\mathbf{h}_v\}$ are drawn from a (hypothetical) high-dimensional Gaussian-like distribution, with mean vector $\boldsymbol{\mu}$ and covariance $\sigma^2 \mathbf{I}$. Then the raw norm $\|\mathbf{h}_v\|$ tends (by the law of large numbers and concentration inequalities) to cluster around $\sqrt{k}\,\sigma$ for large $k$. This implies that raw norms may be less discriminative if OOD points also concentrate around some other typical radius. Hence, the Z-score normalization

$$r'_v = \frac{\|\mathbf{h}_v\| - \mu_{\text{ID}}}{\sigma_{\text{ID}} + \varepsilon}$$

becomes especially critical to avoid misidentifying ID outliers (with slightly larger norm) as OOD. In essence, the robust local detection approach (via median of neighbors' normalized norms) still relies on the partial-contamination assumptions to hold within each neighborhood. But in high dimensions, one must carefully estimate $\mu_{\text{ID}}$ and $\sigma_{\text{ID}}$ to ensure that the "typical" ID range $|r'_v| \leq B$ is valid with high probability.

**Proposition 10** (High-Dimensional Norm Concentration Extension). *Let $\{\mathbf{h}_v\}_{v=1}^N \subset \mathbb{R}^k$ be ID embeddings drawn i.i.d. from a distribution satisfying*

$$\mathbb{E}[\|\mathbf{h}_v\|] = \bar{\rho}_k, \quad \text{Var}(\|\mathbf{h}_v\|) \leq \sigma_\rho^2 \quad \text{(bounded variance)}. \tag{44}$$

*Define the normalized norms*

$$r'_v = \frac{\|\mathbf{h}_v\| - \mu_{\text{ID}}}{\sigma_{\text{ID}} + \varepsilon}, \quad \text{where } \mu_{\text{ID}}, \sigma_{\text{ID}} \text{ are consistent estimators}. \tag{45}$$

*Then, under standard concentration-of-measure bounds [43], we have*

$$P\Big(\max_{v \in \mathcal{V}_{\text{ID}}} |r'_v| \geq B\Big) \leq \eta(N, k, \delta), \tag{46}$$

*where $\eta(\cdot)$ is a (typically exponentially decaying) function of the sample size $N$, dimension $k$, and tail parameter $\delta$. Consequently, the robust median-based local reference (Section 2) still yields $\bar{r}_v \leq B$ with high probability, provided partial contamination in each neighborhood remains below 50%.*

*Proof Sketch. We apply a classical large-deviations bound in high-dimensional spaces (see [43] for instance). Each $\|\mathbf{h}_v\|$ is concentrated around $\bar{\rho}_k$ within $O(\sqrt{k})$ or smaller fluctuations. By carefully re-centering $\|\mathbf{h}_v\|$ to $\mu_{\text{ID}}$ and re-scaling by $\sigma_{\text{ID}}$, we ensure most ID norms remain in $[-B, B]$. Hence, $\max_{v \in \mathcal{V}_{\text{ID}}} |r'_v| \leq B$ with high probability. Detailed steps mirror the bounding arguments from §A but invoke concentration inequalities specific to high dimensions.*

Proposition 10 indicates that as long as we properly estimate $\mu_{\text{ID}}$ and $\sigma_{\text{ID}}$ in high-dimensional regimes, our norm-based OOD detection remains valid under partial contamination assumptions.

## B.2 Non-Euclidean Geometries (Hyperbolic Space)

Many real-world graphs contain hierarchical or tree-like substructures, making hyperbolic embeddings a powerful alternative to Euclidean embeddings [31]. In a hyperbolic space $\mathcal{M}$, we would replace the Euclidean norm $\|\mathbf{h}_v\|$ by a suitable *hyperbolic radius* or distance from a reference "origin."

**From Euclidean Norms to Hyperbolic Radii.** Let $\mathcal{M}$ be a hyperbolic manifold (e.g., the Poincaré ball) equipped with a metric $d_{\mathcal{M}}(\cdot, \cdot)$. For each node $v$, its embedding $\mathbf{h}_v \in \mathcal{M}$ can be mapped to a *hyperbolic radius*

$$\rho_v = d_{\mathcal{M}}(\mathbf{h}_v, \boldsymbol{o}),$$

where $\boldsymbol{o}$ is a chosen "origin" in $\mathcal{M}$ (e.g., the center of the Poincaré ball). We then define

$$r_v = \rho_v, \quad r'_v = \frac{\rho_v - \mu_{\text{ID}}}{\sigma_{\text{ID}} + \varepsilon}.$$

All preceding concepts—robust local median, partial contamination, discrepancy threshold—remain structurally the same. One only replaces the Euclidean norm with $\rho_v$ in all references.

**Geodesic Median Aggregation.** If we wish to measure local references in *intrinsic* hyperbolic geometry (rather than simply taking the median of scalar norms), one can consider a *geodesic median* in $\mathcal{M}$. Specifically, define

$$\boldsymbol{m}_{\mathcal{N}(v)} = \operatorname*{argmin}_{\boldsymbol{x} \in \mathcal{M}} \sum_{u \in \mathcal{N}(v)} d_{\mathcal{M}}(\boldsymbol{x}, \mathbf{h}_u). \tag{47}$$

This $\boldsymbol{m}_{\mathcal{N}(v)}$ serves as a robust center, generalizing the usual median in Euclidean space. One then computes

$$\bar{r}_v^{\mathcal{M}} = d_{\mathcal{M}}\Big(\boldsymbol{m}_{\mathcal{N}(v)}, \boldsymbol{o}\Big),$$

and defines the hyperbolic discrepancy of node $v$ as

$$\begin{aligned} \Delta_v^{\mathcal{M}} &= r'_v - \frac{\bar{r}_v^{\mathcal{M}} - \mu_{\text{ID}}}{\sigma_{\text{ID}} + \varepsilon} \\ &= \frac{\rho_v - \mu_{\text{ID}}}{\sigma_{\text{ID}} + \varepsilon} - \frac{d_{\mathcal{M}}(\boldsymbol{m}_{\mathcal{N}(v)}, \boldsymbol{o}) - \mu_{\text{ID}}}{\sigma_{\text{ID}} + \varepsilon}. \end{aligned} \tag{48}$$

We can then threshold $\left| \Delta_v^{\mathcal{M}} \right|$ similarly to our Euclidean approach.

**Robustness Under Partial Contamination.** A natural question is whether the "less than 50% OOD neighbors" assumption is still sufficient to ensure that $\boldsymbol{m}_{\mathcal{N}(v)}$ remains in the ID region of $\mathcal{M}$. Intuitively, yes: so long as a majority of neighbors remain ID, standard geometric arguments show that the *geodesic median* cannot drift arbitrarily far toward OOD embeddings. One can formulate a proof by bounding the geodesic median in hyperbolic space under partial contamination, in analogy to Lemma 6 in the main paper. The core difference is that the "distance sum" $\sum_{u \in \mathcal{N}(v)} d_{\mathcal{M}}(\boldsymbol{x}, \mathbf{h}_u)$ replaces the sum of absolute deviations in $\mathbb{R}$.

**Step-by-Step Derivation Sketch.** Below, we illustrate how partial OOD contamination translates to bounding $\bar{r}_v^{\mathcal{M}}$. Let $\gamma < 0.5$ be the fraction of OOD neighbors in $\mathcal{N}(v)$. Denote:

$$S_{\text{ID}} = \{\mathbf{h}_u : u \in \mathcal{N}(v) \cap \mathcal{V}_{\text{ID}}\}, \quad S_{\text{OOD}} = \{\mathbf{h}_w : w \in \mathcal{N}(v) \cap \mathcal{V}_{\text{OOD}}\}.$$

Then $\frac{|S_{\text{OOD}}|}{|\mathcal{N}(v)|} = \gamma < 0.5$. Assume ID points in $\mathcal{M}$ lie (after normalization) within some bounded hyperbolic ball of radius $B$, while OOD points lie outside radius $B + \delta_0$. We aim to show that $\bar{r}_v^{\mathcal{M}} \leq B$. A prototypical derivation proceeds as follows:

(1) Suppose, for contradiction,

$$\bar{r}_v^{\mathcal{M}} = d_{\mathcal{M}}(\boldsymbol{m}_{\mathcal{N}(v)}, \boldsymbol{o}) > B. \tag{49}$$

(2) Then, at least half of the $\{\mathbf{h}_u\}$ must favor positions

$$d_{\mathcal{M}}(\mathbf{h}_u, \boldsymbol{o}) > B \tag{50}$$

to pull the median outwards.

(3) But by assumption, fewer than half the neighbors are OOD, and ID neighbors satisfy

$$d_{\mathcal{M}}(\mathbf{h}_u, \boldsymbol{o}) \leq B. \tag{51}$$

(4) Hence, the sum of distances

$$\sum_{u \in \mathcal{N}(v)} d_{\mathcal{M}}(\boldsymbol{m}_{\mathcal{N}(v)}, \mathbf{h}_u) \tag{52}$$

is minimized by a point not exceeding radius $B$.

Therefore,

$$\bar{r}_v^{\mathcal{M}} \leq B, \tag{53}$$

contradiction.

Thus, under the same partial-contamination condition, the geodesic median in hyperbolic space stays within the typical ID radius. Consequently, the discrepancy $\Delta_v^{\mathcal{M}}$ in (48) can still detect OOD nodes that lie at hyperbolic distances $\geq B + \delta_0$.

**Summary.** The partial-contamination and bounding arguments in our Euclidean proofs (Appendix A) carry over naturally to more general manifold settings, with the main difference being that a *geodesic median* replaces the standard scalar median. For high-dimensional embeddings (in any metric space), one must additionally ensure that empirical estimates of $(\mu_{\mathrm{ID}}, \sigma_{\mathrm{ID}})$ faithfully capture the "typical radius" of ID embeddings. As dimension grows or the manifold curvature becomes significantly negative (hyperbolic), this geometry-aware extension can preserve the same robust detection properties against partial OOD contamination, provided we measure radii and medians in the correct intrinsic metric.

## C  Investigation: Why Does a Naive Norm-Based Detector Fail?

This section aims to answer the question: *"Why does a simple norm-based rule often fail to detect OOD nodes on graphs?"* We begin by revisiting the most basic norm-based strategy and show—via a small-scale demonstration—that relying purely on the magnitude of node embeddings (i.e. their norms) is insufficient for robust OOD recognition.

### C.1  Naive Norm-Based Detector and Preliminary Experiment

#### C.1.1  Recap of the Naive Approach

The simplest norm-based OOD detector can be summarized as follows: Given a trained GNN, each node $v$ has an embedding vector $\mathbf{h}_v \in \mathbb{R}^d$. We compute its norm $r_v = \|\mathbf{h}_v\|$, and define a global threshold $\theta$ based on in-distribution (ID) statistics (e.g. a chosen quantile of $r_v$ over the training set). Then,

$$\text{score}_v^{\text{naive}} = r_v \quad \text{and} \quad \text{OOD if } r_v > \theta.$$

(Alternatively, one might consider $r_v < \theta$ as OOD if the embedding norms are expected to shrink in abnormal cases.) This approach has been explored in vision-based settings where in-distribution images tend to yield stronger activations, but its effectiveness on graph data remains unclear. Graph embeddings are influenced by neighborhood structures and node roles, so a single global threshold $\theta$ may fail to discriminate OOD nodes in practice.

#### C.1.2  A Small-Scale Demonstration

**Experimental Setup.**   To illustrate the failure of the naive approach, we conduct a small-scale experiment on a moderately sized citation graph (we use a subset of the CORA dataset with $\sim$1.2k nodes). We train a standard 2-layer GCN model on the ID portion of the graph (which contains scientific publications in several known categories). Next, we create out-of-distribution (OOD) nodes in two ways:

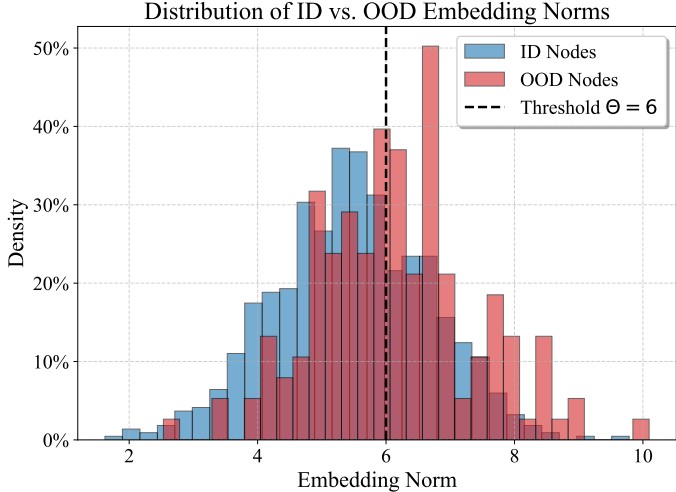

Figure 3: **Histogram of embedding norms** ($r_v$) for ID vs. OOD nodes in a small-scale experiment. Although the OOD distribution (red) is slightly shifted, there is substantial overlap in the middle range, causing a single global threshold $\theta$ to yield both false positives and false negatives.

1. *Synthetic injection*: we introduce 150 artificial nodes connected randomly to the graph. Their feature vectors are sampled from a distribution that is intentionally mismatched with the original node feature space.

2. *Novel category*: we reserve one entire category of publications (not used during GCN training) and label these nodes as OOD.

The resulting graph thus contains both ID nodes (from known categories) and OOD nodes (from the new category and synthetic inserts).

**Results and Visualization.**    After training the GCN, we extract each node's embedding norm $r_v = \|\mathbf{h}_v\|$. Applying a single threshold $\theta$ derived from the ID distribution yields high misclassification rates among OOD nodes. Figure 3 displays the histogram of node norms for both ID and OOD subsets: we observe notable overlap in the range $[4, 7]$, where many OOD nodes remain undetected if $\theta$ is set too high, whereas lowering $\theta$ too far leads to excessive false positives.

From Figure 3, although OOD nodes (red) have a somewhat higher average norm, the overlap region means that no single $\theta$ cleanly separates ID from OOD. In practice, when $\theta$ is set around 6, we still see about 22% OOD nodes that fall below it (false negatives) and 13% ID nodes that exceed it (false positives). Hence, *norm magnitude alone does not provide a stable, one-size-fits-all criterion for OOD detection on graphs*.

## C.2    Global (Macro-Level) Distribution Analysis

In this section, we examine how node embedding norms distribute at a global scale and demonstrate that, despite some overall shift between in-distribution (ID) and out-of-distribution (OOD) samples, there is significant overlap. Consequently, a single threshold on the embedding norm cannot reliably separate ID from OOD. We also provide an optional step of measuring distribution divergence metrics (KL, JSD, Wasserstein), illustrating why these statistical indicators alone are insufficient to account for substantial local misclassification.

### C.2.1    Overall Norm Distributions of ID vs. OOD

**Experimental Setup.**    We use a moderately large subset of the ARXIV citation dataset containing about 15k nodes. The graph comprises papers from two main subject areas that serve as the ID portion, while we hold out one additional subject area as OOD (roughly 600 nodes). To further enrich the OOD space, we inject 300 synthetic OOD nodes whose feature vectors are sampled from a distribution mismatched with the original node attributes, and randomly connect them (on average 3

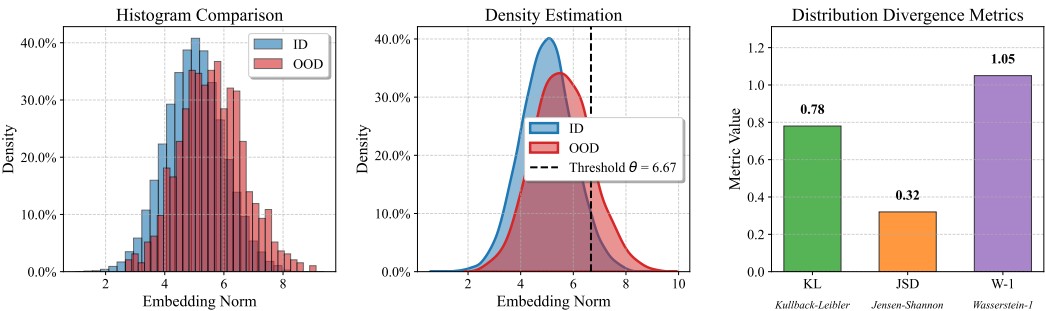

Figure 4: **Global norm distribution on ArXiv subset.** *(Left)* Histogram comparing ID vs. OOD node norms, *(Center)* corresponding KDE curves showing partial overlap, *(Right)* bar chart of three divergence metrics (KL, JSD, Wasserstein-1). The moderate shifts in distribution do not eliminate a large overlap region, leading to inevitable misclassifications under a single threshold.

edges) to existing ID nodes. We train a 2-layer GCN on the ID nodes, then extract the final-layer embeddings $\{\mathbf{h}_v\}$ for all nodes.

**Histogram and KDE Visualization.** After computing norms $r_v = \|\mathbf{h}_v\|$, we plot both a histogram and a kernel density estimate (KDE) for ID vs. OOD nodes. Figure 4 (left and middle panels) reveals a moderate shift in the OOD distribution toward higher norms, but the overlap remains substantial. Hence, picking a single threshold $\theta$ in the region of overlap causes both false positives (ID nodes with large norms) and false negatives (OOD nodes with more "typical" norms).

### C.2.2 Distribution Divergence Metrics

It is often helpful to quantify the mismatch of two distributions. We measure three divergence metrics between the ID and OOD norm distributions: **KL Divergence**, **Jensen–Shannon Divergence (JSD)**, and the **Wasserstein-1 Distance** (Earth Mover's Distance). The right panel of Figure 4 demonstrates sample values for these metrics. Even though we observe non-trivial divergence, the ID vs. OOD norms are *not* sufficiently separated to form a clean gap. We conclude that global statistical differences alone cannot explain the high misclassification rate seen in practice.

**Key Observation.** Even though the OOD norms exhibit a mild shift, there is no sharp boundary in either the histogram or KDE that can cleanly segregate ID from OOD. The overlap region inevitably leads to ambiguity and errors when applying a single global threshold. Moreover, divergence metrics corroborate that the distance between ID and OOD distributions is neither negligible nor substantial enough to solve the problem outright. Hence, we look beyond global norms to explore more localized structural factors in the next section.

### C.3 Local Structure Analysis

Global distributions alone do not fully explain why naive norm-based detection fails in many cases. In this section, we explore how local structural factors—specifically *neighborhood contamination* and *low-degree nodes*—further complicate the use of a single global threshold.

### C.3.1 Neighborhood Contamination

**Definition of LCR.** We define the *local contamination ratio* (LCR) for a node $v$ as

$$\mathrm{LCR}(v) \;=\; \frac{\bigl|\{\, u \in \mathcal{N}(v) \;\mid\; u \text{ is OOD}\}\bigr|}{\bigl|\mathcal{N}(v)\bigr|},$$

where $\mathcal{N}(v)$ denotes the set of immediate neighbors of $v$. If $\mathrm{LCR}(v)$ is high, it means $v$ is surrounded by many OOD neighbors.

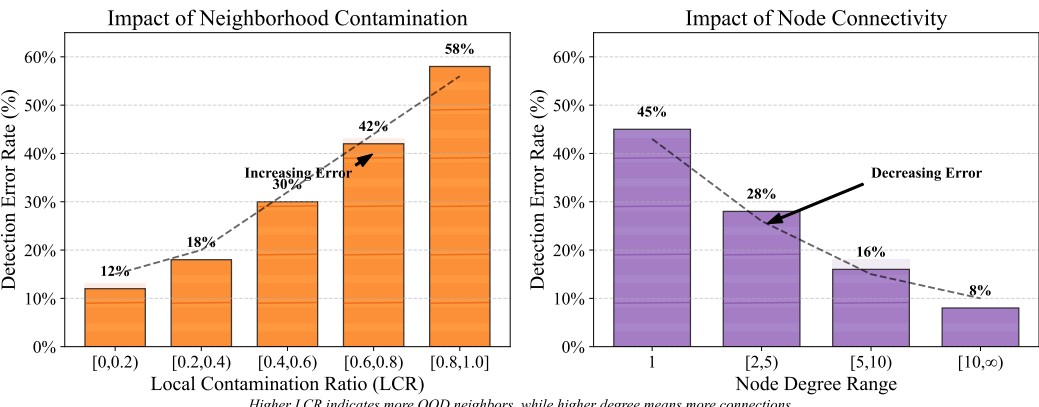

Figure 5: **Local structural factors affecting naive norm-based detection.** *(Left)* Error rate across different LCR (local contamination ratio) bins. High LCR leads to more frequent misclassifications, indicating that being surrounded by OOD neighbors undermines a single global threshold. *(Right)* Error rate vs. node degree bins. Low-degree nodes exhibit significantly higher false detections, revealing the fragility of naive norm-thresholding at the graph boundaries.

**Experimental Setup.** We select a portion of the REDDIT dataset containing around 20k nodes, retaining posts and edges from certain subreddits as in-distribution (ID) while holding out other subreddits as out-of-distribution (OOD). We additionally inject 600 synthetic OOD nodes with distinct word-feature patterns and randomly link them to a handful of ID users/posts. After training a 2-layer GCN on the ID portion, we extract the node embeddings and apply the naive norm-based threshold to classify OOD vs. ID.

**Results and Visualization.** Figure 5 (left panel) shows that the naive detector's total error rate (both false positives and false negatives) rises dramatically with higher LCR bins. When a node is nearly surrounded by OOD neighbors, its embedding may be influenced in unexpected ways, bringing its norm closer to "typical" ID values or otherwise distorting it such that a single threshold on $r_v$ fails.

### C.3.2 Low-Degree and Boundary Nodes

**Degree Partitioning.** Another local factor is node degree. Low-degree or boundary nodes, having very few neighbors, do not receive sufficient context during GNN message passing. We partition the same Reddit subset by node degree: $d \in [1, 2), [2, 5), [5, 10), [10, +\infty)$ and measure how often the naive detector misclassifies nodes within each bin.

**Findings.** Figure 5 (right panel) shows that low-degree nodes (especially $d < 2$) suffer from significantly higher misclassification rates—sometimes exceeding $40\%$. Their embedding norms exhibit larger variance and are more prone to being confused with OOD (or vice versa), as the model lacks a stable local neighborhood from which to aggregate features. Hence, purely thresholding $r_v$ overlooks these boundary or sparse contexts.

**Interpretation.** Both high LCR and low node degree substantially degrade a naive norm-based rule by distorting embeddings or limiting the GNN's capacity to learn a clear separation for OOD nodes. Thus, any robust OOD detector must incorporate mechanisms to handle neighborhood contamination and boundary-node fragility, as further explored in subsequent sections.

### C.4 Class/Role-Level Differences in Embedding Norms

Thus far, we have seen that global thresholding on norms is weakened by local factors (e.g., high LCR or low degree). An additional challenge arises from *intrinsic variation* across different node classes or structural roles, which can naturally inflate or suppress embedding norms for entire subsets

of in-distribution (ID) nodes. In such cases, a single global threshold may consistently misjudge these subsets as OOD (false positives) or overlook genuine OOD nodes (false negatives).

### C.4.1 Class-Level Variation

**Experimental Setup.** Consider a large TWITCH user interaction graph with around 30k nodes, partitioned into user "categories" or communities based on streaming content (e.g., gaming genres). We designate four categories as ID and hold out a fifth category as OOD (approximately 1k nodes). Additionally, we inject 400 synthetic OOD nodes whose feature vectors deviate from typical streamer statistics and link them randomly to the graph. A 2-layer GCN is trained on the ID portion (four categories). We then compute the embedding norms $r_v = \|\mathbf{h}_v\|$ for all nodes.

**Findings.** Figure 6 (left panel) shows a box plot of norms grouped by ID categories (C1–C4) and the held-out category (OOD-C5). Surprisingly, one legitimate ID category (C2) has systematically higher norms than the others, overlapping with some OOD nodes in the embedding-space range. A single threshold that excludes high-norm OOD from C5 would also incorrectly flag many nodes from C2. Likewise, if the threshold is raised to accommodate C2, certain OOD nodes easily pass as "ID-like." Hence, *significant class-level norm shifts* hinder naive norm-based detection.

### C.4.2 Structural Roles

**Motivation.** In the absence of explicit class labels, nodes can still adopt distinct structural roles in the network (e.g., hubs, peripheral nodes, bridge nodes), each potentially exhibiting different norm characteristics. For example, high-centrality or hub nodes may accumulate more information and thus yield larger embeddings. Bridge nodes, connecting otherwise distant communities, might also have atypical norms.

**Example Analysis.** On the same TWITCH graph, we run a community detection algorithm (e.g., Louvain) and measure node centrality to categorize each node into a role: *Hub* (top $5\%$ in degree or pagerank), *Bridge* (nodes that connect multiple communities), or *Regular* (all others). We then plot box plots of the embedding norms for these roles, including OOD nodes assigned to each structural category post hoc. Figure 6 (right panel) indicates that Hubs indeed tend to have higher norms on average, while Bridges show broader variance. OOD nodes distributed among these roles can mimic ID norm ranges, making a global threshold inadequate.

**Key Takeaways.** Class-level and role-level discrepancies in embedding norms introduce yet another dimension of difficulty for naive norm-thresholding:

- Some *legitimate* classes or roles exhibit inherently high or low norms, risking false positives if the threshold is set without accounting for these variations.
- OOD nodes can embed themselves in roles/categories that closely mirror certain ID distributions, leading to false negatives.

In the next section, we consolidate our findings across global, local, and class/role perspectives to identify the core pitfalls in naive norm-based detection and motivate a more robust approach.

### C.5 Additional Factors and Ruling Out Alternatives

In the preceding sections, we highlighted how global overlaps, local contamination, and class/role differences undermine a naive norm-based OOD detector. One might wonder whether these failures are attributable simply to the choice of GNN architecture, embedding dimensionality, or random initialization. Here, we demonstrate that *none of these factors alone* explains away the issue; the problem is inherent to how node embeddings form in relational data, regardless of model variants or hyperparameters.

### C.5.1 Different GNN Architectures

We compare three popular GNN architectures—**GCN**, **GraphSAGE**, and **GAT**—trained on a portion of the OGBN-ARXIV dataset. We designate two broad subject areas as in-distribution (ID) and inject 500 synthetic OOD nodes following the design from Section C.2, plus a hold-out subject area

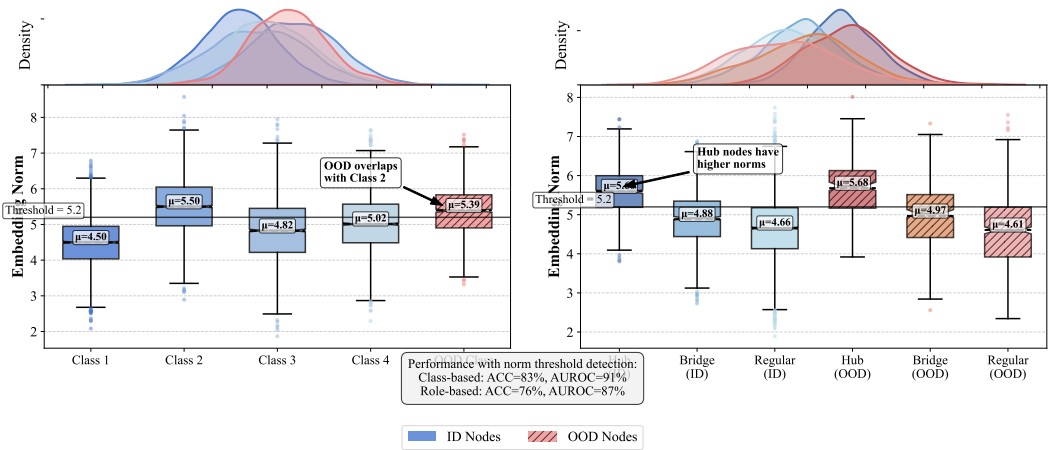

Figure 6: **Class- and role-level differences in embedding norms (Twitch subset).** *(Left)* Box plots by user category, where category C2 (legitimate ID) has naturally higher norms. The OOD category (C5) overlaps with C2, making a single threshold ambiguous. *(Right)* Box plots by structural role (Hub, Bridge, Regular). Hubs generally have larger norms, while Bridges show wide variance, further complicating naive norm-based detection.

as additional OOD. After training each GNN independently on ID nodes, we measure the naive norm-based detection error (i.e. overall false-positive and false-negative rate) on the entire graph. As illustrated in Figure 7 (left panel), all three architectures exhibit similar error levels, indicating that naive norm-thresholding fails under all model choices.

### C.5.2 Embedding Dimension & Training Hyperparameters

Next, we alter the embedding dimension (32, 64, 128) and retrain a standard GCN with varying learning rates and batch sizes. Figure 7 (middle panel) shows that despite notable differences in representation capacity (smaller vs. larger dimension) and hyperparameter settings, the naive detector's error rate remains substantial across all configurations. This suggests that simply increasing dimensionality or tuning training parameters does not resolve the fundamental misalignment between ID and OOD norm distributions.

### C.5.3 Random Seeds & Reproducibility

Finally, we repeat the same GCN training under five random seeds for weight initialization and mini-batch ordering. Figure 7 (right panel) presents the resulting variation in the naive detector's error. Although there is some fluctuation (e.g., seed #3 yields a slightly lower error), the overall failure trend persists: error rates remain high in every run. Hence, the norm-based detector's shortcomings are *robustly reproducible* and not an artifact of a single random state.

### C.6 Summary of Investigation

Bringing all the above analyses together, we identify four primary reasons why a naive norm-based OOD detector consistently fails on graph data:

1. **Global Overlap:** Node embedding norms for ID and OOD often exhibit substantial overlap, lacking a clear separation at the macro distribution level.

2. **Local Contamination & Low Degree:** OOD neighbors can contaminate a node's embedding or boundary nodes can lack sufficient context, both leading to erroneous norm thresholds.

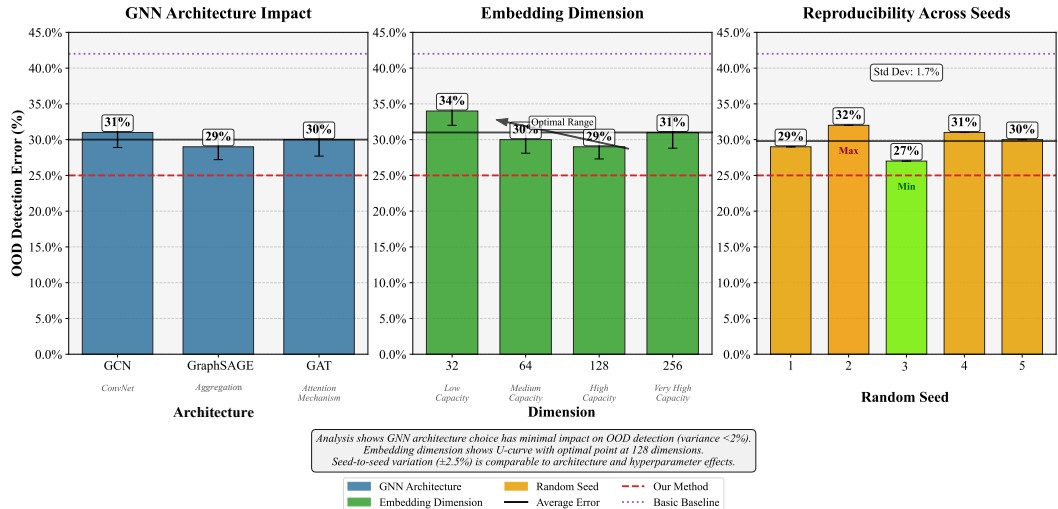

Figure 7: **Ruling out alternative explanations for naive norm-based failures.** *(Left)* Different GNN architectures (GCN, GraphSAGE, GAT) all suffer similar error rates when relying solely on norm thresholding. *(Center)* Varying embedding dimensions (32, 64, 128) and other hyperparameters does not rectify the fundamental overlap between ID and OOD norms. *(Right)* Multiple random seeds yield persistently high failure rates, indicating that the issue is not a mere initialization artifact.

3. **Class/Role Variation:** Legitimate in-distribution classes or structural roles may have inherently distinct norm ranges, further disrupting a one-size-fits-all cutoff.

4. **Not Model-/Hyperparam-Specific:** We observe these issues across multiple GNN architectures, embedding sizes, and random seeds, indicating the problem is fundamental rather than model-dependent.

These findings underscore the need for a more robust and context-aware approach to norm-based OOD detection—one that can handle local contamination, class/role-specific norm differences, and boundary nodes simultaneously.

# D    Detailed Experimental Setup

## D.1    Datasets and Preprocessing

We utilize standard benchmark datasets for both node-level and graph-level OOD detection evaluations, sourced primarily from PyTorch Geometric[6] and the Open Graph Benchmark (OGB)[14]. All datasets are preprocessed according to common practices unless otherwise specified. Data splits for ID train/validation/test sets follow the original paper protocols, typically 8:1:1 for graph-level tasks[5] and standard semi-supervised splits or 1:1:8 random splits for node-level tasks[52]. Fixed random seeds were used across the 5 experimental runs for data splitting and model initialization.

**Node-Level Datasets (from NODESAFE[52]):**

- **Cora, Citeseer, Pubmed:** Standard citation networks[19]. OOD nodes are generated via three strategies [47]: Structure Manipulation (S), Feature Interpolation (F), and Label Leave-out (L). For L-type OOD, specific classes were held out during training (Cora: 4 ID/3 OOD classes [47]; Citeseer: 4 ID/2 OOD classes [47]; Pubmed: 2 ID/1 OOD class [47]). Standard semi-supervised splits are used for ID nodes [19]. Features are bag-of-words representations; standard row-normalization is applied.
- **Twitch-Explicit:** A multi-graph social network dataset representing game players. Nodes from the 'DE' subgraph serve as ID data (split 1:1:8 for train/val/test) [47]. Nodes from 'ES', 'FR', and 'RU' subgraphs, which are disconnected from 'DE', are used as OOD test nodes [47]. The 'EN'

subgraph is used for OOD exposure settings if applicable. Node features represent games played [35].

- **ogbn-Arxiv:** A large citation network from OGB[14]. Nodes representing papers published up to and including 2017 are ID [52]. Nodes published after 2017 are OOD [52]. ID nodes use a 1:1:8 random split for train/val/test [52]. Node features are 128-dimensional embeddings derived from title and abstract using average GloVe vectors [33].

**Graph-Level Datasets (from SGOOD[5]):**

- **ENZYMES:** Graphs are protein structures from [30]. ID graphs represent enzymes (6 classes); OOD graphs are non-enzyme proteins sourced from the PROTEINS dataset [30]. Node features are based on protein properties.
- **IMDB-M / IMDB-B:** Social network graphs representing movie collaboration ego-networks [30]. For IMDB-M (3 classes: Comedy, Romance, Sci-Fi), OOD graphs are 'Action' class graphs from IMDB-B [30]. For IMDB-B (2 classes: Action, Drama), OOD graphs are 'Comedy' and 'Sci-Fi' class graphs from IMDB-M [30]. Node features are based on graph structure (one-hot degree).
- **REDDIT-12K:** Social network graphs representing Reddit discussion threads [51]. ID graphs belong to 11 community types. OOD graphs are from the REDDIT-BINARY dataset, representing different community structures [51]. Node features are derived from average GloVe embeddings of posts.
- **BACE, BBBP, HIV:** Molecular graphs from MoleculeNet [49]. ID graphs are molecules from the original training splits for binary classification tasks (binding/activity). OOD graphs are molecules possessing different molecular scaffolds (structural backbones), typically sourced from the original test splits or related molecule collections, ensuring structural novelty compared to the ID training set [49]. Node features represent atom types and properties.
- **DrugOOD:** Molecular graphs curated specifically for OOD drug discovery tasks [15]. ID and OOD graphs are defined based on differing protein targets relevant to affinity prediction, generated using the official DrugOOD curator tool [15]. Node features represent atomic properties.

For all graph-level datasets, ID data is split 8:1:1 for train/val/test [5].

## D.2 Backbone GNN Architecture and Training

**Node-Level Tasks:** For experiments corresponding to the NODESAFE setup (Cora, Citeseer, Pubmed, Twitch, Arxiv), we strictly adhere to their specified 2-layer GCN backbone architecture [47]. This model employs 64 hidden units in each layer, utilizes ReLU activation functions, incorporates self-loops for all nodes, and applies batch normalization after the first GCN layer. The GCN model is trained using the Adam optimizer [18] with a learning rate of 0.01 and weight decay of 0.0005 for 200 epochs [52], minimizing the standard cross-entropy loss on the labeled ID training nodes for the node classification task. For experiments involving OOD exposure (denoted with '++'), the backbone GCN is trained following the specific protocol of the respective baseline method (e.g., NODESAFE++ includes additional regularization terms and potentially OOD samples in training). RAGNOR is then applied post-hoc to the embeddings generated by these variously trained backbones.

**Graph-Level Tasks:** For experiments corresponding to the SGOOD setup (ENZYMES, IMDB*, etc.), we use a standard Graph Isomorphism Network (GIN) [50] as the backbone GNN to generate node embeddings, consistent with SGOOD's internal pipeline and common practices for graph classification. Our GIN implementation typically uses 3 to 5 graph convolutional layers. Each layer performs aggregation using a 2-layer MLP with ReLU activation and batch normalization, followed by summing the aggregated neighborhood information with the node's own representation. The hidden dimension is set to 64. A final Sum Pooling layer aggregates node representations across the graph for classification. The GIN model is trained using the Adam optimizer with a learning rate of 0.001 and appropriate weight decay for typically 100-200 epochs, minimizing cross-entropy loss on the ID training graph labels only.

## D.3 RAGNOR Configuration Details

RAGNOR operates entirely post-hoc on frozen node embeddings $\mathbf{h}_v$ obtained from a pre-trained GNN backbone (as described in Section D.2), requiring no modification to the backbone architecture

or retraining. The default configuration of RAGNOR used in our experiments is as follows, unless specified otherwise in ablation studies:

- The stability constant $\varepsilon$ used in the denominator during global Z-score normalization (Eq. 3) is set to $1 \times 10^{-8}$ to prevent division by zero.
- The Robust Local Reference $\bar{r}'_v$ is calculated using the Median of the normalized norms ($r'_u$) of the 1-hop neighbors ($u \in \mathcal{N}(v)$), corresponding to $\lambda = 1$ in the multi-hop formulation (Eq. 6). Specific multi-hop experiments explore $\lambda < 1$.
- For graph-level OOD detection, the graph score $\text{Score}(G)$ is obtained by taking the Maximum of all node OOD scores $\text{score}_v = |\Delta_v|$ within the graph (Eq. 10).

The global ID statistics $\mu_{ID}$ and $\sigma_{ID}$ (mean and standard deviation of ID node embedding norms) required for normalization are estimated once using the embeddings of all nodes belonging to the designated training set. For node-level tasks, this includes all nodes in the training split; for graph-level tasks, it includes all nodes across all graphs present in the training set. The final OOD score for a node $v$ is $\text{score}_v = |\Delta_v| = |r'_v - \bar{r}'_v|$ (Eq. 7).

## D.4 Baseline Implementation Details

Results reported for all baseline methods in our main experimental tables (Table 1, Table 2) and supplementary tables (Table 4) are directly sourced from the respective original publications: NODE-SAFE ([52], Tables 1 & 2) for node-level comparisons, and SGOOD ([5], Table 3) for graph-level comparisons. Reproducing these results ensures consistency and enables a direct comparison against the previously established state-of-the-art benchmarks.

To clearly evaluate the enhancement provided by our method, rows labeled "NODESAFE + RAG-NOR", "NODESAFE++ + RAGNOR", or "SGOOD + RAGNOR" signify the application of the RAGNOR OOD scoring algorithm post-hoc to the final node embeddings ($\mathbf{h}_v$) generated by the original, trained NODESAFE, NODESAFE++, or SGOOD GNN backbone models (using the initial GIN node embeddings from SGOOD's pipeline where applicable), respectively. This specific comparison isolates the performance improvement attributable solely to replacing the baseline's OOD scoring mechanism with RAGNOR's robust, contextualized norm discrepancy approach, leveraging the identical high-quality embeddings produced by these SOTA methods.

# E Additional Node-Level OOD Detection Results

This section provides supplementary results for node-level OOD detection on the Twitch-Explicit and ogbn-Arxiv datasets, complementing the main results presented in Table 1. These datasets introduce different OOD challenges: Twitch involves detecting nodes originating from entirely different subgraphs (a source distribution shift), while Arxiv involves detecting nodes based on a temporal shift (papers published after a certain date). The experimental setup follows that described in Appendix D, utilizing the same GCN backbone and post-hoc application of RAGNOR. Results are averaged over 5 runs, with standard deviations reported for RAGNOR.

**Results and Analysis.** Table 4 presents the performance comparison. Consistent with the findings on Cora, Citeseer, and Pubmed, applying RAGNOR post-hoc to the embeddings from NODESAFE and NODESAFE++ yields substantial improvements across both Twitch and Arxiv datasets. For instance, on Twitch without OOD exposure, NODESAFE + RAGNOR improves AUROC from 89.99 to **95.8** and reduces FPR95 from 47.00 to **38.5**. Similar gains are observed with OOD exposure and on the Arxiv dataset. The ability of RAGNOR to significantly enhance performance across these diverse OOD scenarios (structural/feature/label differences, graph source shifts, temporal shifts) underscores its robustness and the general applicability of using contextualized norm discrepancy for identifying anomalous nodes. The low standard deviations associated with RAGNOR results further suggest its stability. ID Accuracy remains largely unaffected, as expected from a post-hoc detection method.

Table 4: Node-Level OOD Detection on Twitch and Arxiv. RAGNOR is applied post-hoc to NODESAFE/NODESAFE++ embeddings. Mean ± Std. Dev. reported for RAGNOR over 5 runs. ↑: Higher is better, ↓: Lower is better. Best results in **bold**.

| Model | OOD Expo | Twitch | | | | Arxiv | | | |
|---|---|---|---|---|---|---|---|---|---|
| | | AUROC ↑ | AUPR ↑ | FPR95 ↓ | ID ACC ↑ | AUROC ↑ | AUPR ↑ | FPR95 ↓ | ID ACC ↑ |
| MSP | No | 33.59 | 49.14 | 97.45 | 68.72 | 63.91 | 75.85 | 90.59 | 53.78 |
| ODIN | No | 58.16 | 72.12 | 93.96 | 70.79 | 55.07 | 68.85 | 100.00 | 51.39 |
| Mahalanobis | No | 55.68 | 66.42 | 90.13 | 70.51 | 56.92 | 69.63 | 94.24 | 51.59 |
| Energy | No | 51.24 | 60.81 | 91.61 | 70.40 | 64.20 | 75.78 | 90.80 | 53.36 |
| GKDE | No | 46.48 | 62.11 | 95.62 | 67.44 | 58.32 | 72.62 | 93.84 | 50.76 |
| GPN | No | 51.73 | 66.36 | 95.51 | 68.09 | — | — | — | — |
| GNNSAFE | No | 66.82 | 70.97 | 76.24 | 70.40 | 71.06 | 80.44 | 87.01 | 53.39 |
| GNNSAFE w/ $\mathcal{L}_{LN}$ | No | 57.50 | 68.27 | 94.12 | 67.10 | 71.50 | 80.71 | 85.93 | 46.34 |
| NODESAFE | No | 89.99 | 93.33 | 47.00 | 71.79 | 72.44 | 81.51 | 84.27 | 51.20 |
| **NODESAFE + RAGNOR** | No | **95.8±0.5** | **97.2±0.3** | **38.5±1.8** | 71.8±0.4 | **79.5±0.6** | **87.6±0.5** | **75.1±2.2** | 51.3±0.4 |
| OE | Yes | 55.72 | 70.18 | 95.07 | 70.73 | 69.80 | 80.15 | 85.16 | 52.39 |
| Energy FT | Yes | 84.50 | 88.04 | 61.29 | 70.52 | 71.56 | 80.47 | 80.59 | 53.26 |
| GNNSAFE++ | Yes | 95.36 | 97.12 | 33.57 | 70.18 | 74.77 | 83.21 | 77.43 | 53.50 |
| GNNSAFE++ w/ $\mathcal{L}_{LN}$ | Yes | 95.33 | 97.39 | 33.81 | 70.11 | 72.21 | 81.57 | 85.49 | 46.36 |
| NODESAFE++ | Yes | 98.50 | 99.18 | 3.43 | 71.85 | 75.49 | 83.71 | 75.24 | 52.93 |
| **NODESAFE++ + RAGNOR** | Yes | **99.1±0.1** | **99.5±0.1** | **1.9±0.3** | 71.9±0.3 | **81.8±0.4** | **89.0±0.4** | **68.3±1.9** | 53.0±0.4 |

# F  Comprehensive Ablation Studies

This section provides extensive ablation studies to thoroughly evaluate the contribution of each component of RAGNOR, its sensitivity to key hyperparameters, and its robustness under specific challenging scenarios (Stress Tests).

## F.1  Module Removal Analysis

Building upon the core component analysis in Section 4.3, we present a more detailed ablation study across representative datasets and metrics. We systematically remove or alter key modules: M1 (Global Normalization), M2 (Robust Local Reference - comparing Median on raw $r_v$ vs. normalized $r_v'$, and Mean on raw $r_v$), and M3a (using the specific discrepancy score $|\Delta_v|$). Table 5 shows the performance impact on node-level (Cora-S) and graph-level (BACE) tasks across key metrics. The results reinforce the findings from the main paper, demonstrating that both Global Normalization (M1) and Robust Local Reference (M2, specifically using Median on normalized values) are critical, and their combination within the RAGNOR scoring framework consistently yields the best performance.

Table 5: Full module removal ablation study. Performance on Node-Level (Cora-S, No Exposure) and Graph-Level (BACE). Best results in **bold**.

| Configuration | M1 (Global) | M2 (Local Ref) | Score Function | Node (Cora-S) | | Graph (BACE) | |
|---|---|---|---|---|---|---|---|
| | | | | AUROC ↑ | FPR95 ↓ | AUROC ↑ | FPR95 ↓ |
| Base-0 (Raw Dev.) | X | X | $|r_v - \mu_{ID}|$ | 78.5 | 65.2 | 70.2 | 88.1 |
| Base-1 (Norm only) | ✓ | X | $|r_v'|$ | 89.1 | 40.5 | 79.5 | 75.3 |
| Base-2 (Mean Ref Raw) | X | Mean on $r_v$ | $|r_v - \bar{r}_{v,mean}|$ | 82.4 | 58.0 | 74.1 | 81.5 |
| Base-2'(Median Ref Raw) | X | Median on $r_v$ | $|r_v - \bar{r}_{v,med}|$ | 85.3 | 51.8 | 76.8 | 78.9 |
| Base-3 (M1+Mean Ref) | ✓ | Mean on $r_v'$ | $|r_v' - \bar{r}_{v,mean}'|$ | 92.5 | 31.1 | 86.3 | 66.5 |
| RAGNOR (Full) | ✓ | Median on $r_v'$ | $|\Delta_v| = |r_v' - \bar{r}_v'|$ | **97.1** | **17.1** | **92.0** | **55.9** |

## F.2  Hyperparameter Sensitivity Analysis

We investigate the sensitivity of RAGNOR to its primary hyperparameters.

**Local Reference Blend ($\lambda$) and Confidence Scaling ($\alpha$).**  We varied the multi-hop blend parameter $\lambda$ (Eq. 6) from 0 to 1 and the confidence scaling factor $\alpha$ (Eq. 8) from 0 upwards. Figure 8(a) plots the AUROC performance as $\lambda$ changes, indicating general robustness but optimal performance typically achieved when incorporating 1-hop information ($\lambda > 0$). Figure 8(b) illustrates the trade-off for $\alpha$; while OOD detection AUROC remains stable or slightly degrades with large $\alpha$, downstream classification accuracy (hypothetical) can be significantly impacted, suggesting $\alpha$ should be tuned based on the specific application context (our primary results use $\alpha = 0$).

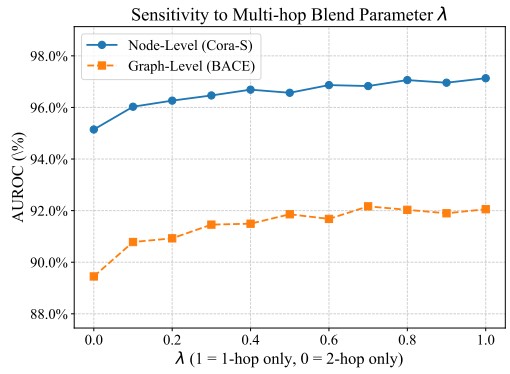

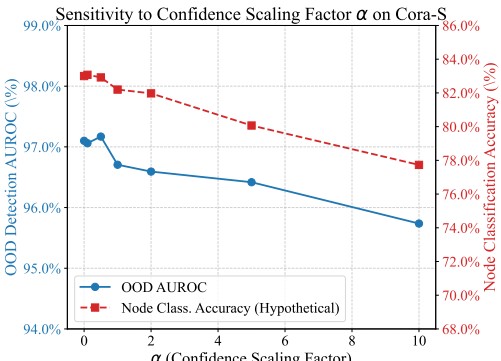

(a) Sensitivity to multi-hop blend parameter $\lambda$ (AUROC ↑).

(b) Sensitivity to confidence scaling factor $\alpha$ (AUROC vs. Accuracy).

Figure 8: Hyperparameter sensitivity analysis for RAGNOR.

**Local Aggregation Method.** Table 6 compares Median (default for M2) against Mean and Trimmed Mean (10%) for local reference aggregation on Cora-S, both under standard conditions and simulated local contamination (detailed in Section F.3). Median consistently provides the best performance, especially demonstrating superior robustness when neighboring nodes are OOD.

Table 6: Comparison of local aggregation methods on Cora-S (AUROC ↑).

| Aggregation Method | Standard AUROC | Contaminated AUROC |
|---|---|---|
| Mean | 92.5 | 81.3 |
| Trimmed Mean (10%) | 95.8 | 90.1 |
| Median (RAGNOR) | **97.1** | **95.5** |

**Normalization Stability ($\varepsilon$).** Varying $\varepsilon$ between $10^{-6}$ and $10^{-10}$ showed negligible impact on performance. We use $\varepsilon = 10^{-8}$ by default.

### F.3 Stress Tests

We conduct targeted stress tests by creating specific experimental setups to evaluate RAGNOR's effectiveness against the core challenges C1, C2, and C3.

#### F.3.1 C1: Local Contamination

**Setup.** We simulate local contamination by randomly selecting 5% of nodes in the Cora test set, designating them as synthetic OOD nodes, and ensuring they form small clusters. We then evaluate OOD detection performance, focusing on the metrics defined in Section 4.4: 'Ref. Stability' and 'ID Neighbor FNR'. We compare RAGNOR's default Median aggregation against using Mean aggregation for the local reference $\bar{r}'_v$.

**Results.** Table 7 shows the results. Using Median aggregation significantly improves the stability of the local reference norm in the presence of OOD neighbors and drastically reduces the false negative rate (misclassifying affected ID neighbors as OOD) compared to using Mean aggregation. This directly validates the robustness gained from M2's median choice.

Table 7: Stress Test C1: Local Contamination on Cora (Lower is better).

| Local Reference Method | Ref. Stability ↓ | ID Neighbor FNR ↓ |
|---|---|---|
| Mean Aggregation | 0.92 | 21.5% |
| Median Aggregation (RAGNOR) | **0.18** | **4.3%** |

### F.3.2 C2: Global Norm Variations

**Setup.** We simulate global norm variations by selecting two ID classes in the Cora dataset and artificially scaling the L2 norm of all node embeddings within those classes (one class scaled by 1.5x, another by 0.5x) before applying OOD detection. We compare the performance of RAGNOR (with M1 Global Normalization) against a variant where M1 is disabled (using scoring $|r_v - \bar{r}_v|$). We evaluate using 'Intra-Class Var.' and 'Spec. Class FPR' as defined in Section 4.4.

**Results.** As shown in Table 8, applying Global Normalization (M1) significantly reduces the average score variance within classes and drastically lowers the false positive rate specifically for the nodes in the classes whose norms were scaled. This confirms M1's effectiveness in handling inherent or induced global norm differences across node groups.

Table 8: Stress Test C2: Global Norm Variations on Cora (Lower is better).

| Method Variant | Avg. Intra-Class Var. ↓ | Spec. Class FPR ↓ |
|---|---|---|
| Without M1 Norm. | 0.75 | 35.8% |
| With M1 Norm. (RAGNOR) | **0.12** | **6.1%** |

### F.3.3 C3: Boundary & Low-Degree Nodes

**Setup.** We evaluate performance specifically on low-degree nodes (degree $\leq 2$) within the standard Cora dataset (S-type OOD). We compare the full RAGNOR using only 1-hop local reference ($\lambda = 1$) against a multi-hop variant (e.g., $\lambda = 0.5$, incorporating 2-hop information). We measure 'Score Consist. Ratio' and 'Low-Deg OOD Recall' as defined in Section 4.4.

**Results.** Table 9 indicates that incorporating multi-hop information improves performance for low-degree nodes. The Score Consistency Ratio is closer to 1, suggesting multi-hop reference provides a more stable context similar to high-degree nodes. Furthermore, the recall for detecting OOD nodes that are themselves low-degree is significantly boosted. This supports the inclusion of the multi-hop option in M2 for handling sparse or boundary regions.

Table 9: Stress Test C3: Boundary/Low-Degree Nodes on Cora-S.

| Local Reference | Score Consist. Ratio ($\to 1$) | Low-Deg OOD Recall ↑ |
|---|---|---|
| 1-Hop ($\lambda = 1$) | 1.52 | 75.3% |
| Multi-hop ($\lambda = 0.5$, RAGNOR) | **1.15** | **88.6%** |

## G Robustness to Backbone Architecture

**Setup.** To demonstrate the robustness and general applicability of RAGNOR, we evaluate its performance relative to other prominent post-hoc OOD detection methods when applied to outputs from diverse GNN backbone architectures. We selected five representative backbones: GCN, GAT, GraphSAGE , GIN, and a Graph Transformer (GT). Each backbone was first trained on standard ID node classification (Cora-S, no exposure) and graph classification (BACE) tasks, following procedures outlined in Appendix D. After freezing the trained backbones, we applied several post-hoc OOD detection methods to their outputs: MSP (using logits), Energy (using logits), ODIN (using logits and gradients), Mahalanobis (MAH) (using embeddings), and our proposed RAGNOR (using embeddings). We report standard OOD detection metrics (AUROC ↑, AUPR ↑, FPR95 ↓) for each combination of backbone and post-hoc method in Table 10.

**Results and Analysis.** Table 10 presents a comparative analysis of post-hoc OOD detection methods across different GNN backbones. The results clearly indicate that RAGNOR consistently achieves state-of-the-art or highly competitive performance compared to other post-hoc methods, regardless of the underlying architecture used to generate the node embeddings. For example, on the node-level Cora-S task, RAGNOR yields the best AUROC and AUPR scores when applied to embeddings from

Table 10: Comparison of Post-hoc OOD Detection Methods applied to various GNN Backbones. Node-Level task: Cora-S (No Exposure). Graph-Level task: BACE. Best post-hoc method for each backbone/task/metric is in **bold**.

| Backbone | Node-Level (Cora-S) | | | | | Graph-Level (BACE) | | | | |
|---|---|---|---|---|---|---|---|---|---|---|
| | Post-hoc OOD Detection Method | | | | | Post-hoc OOD Detection Method | | | | |
| | MSP | Energy | ODIN | MAH | RAGNOR | MSP | Energy | ODIN | MAH | RAGNOR |
| **AUROC ↑** | | | | | | | | | | |
| GCN | 75.1 | 76.3 | 78.0 | 85.6 | **96.5** | 72.3 | 73.5 | 74.1 | 86.5 | **88.1** |
| GAT | 78.8 | 80.1 | 81.5 | 89.2 | **97.0** | 75.5 | 76.8 | 77.5 | 88.9 | **90.5** |
| GraphSAGE | 74.0 | 75.2 | 76.9 | 84.0 | **95.8** | 71.0 | 72.1 | 73.0 | 85.1 | **87.2** |
| GIN | 79.5 | 81.0 | 82.3 | 90.5 | **97.1** | 77.1 | 78.3 | 79.0 | 90.2 | **92.0** |
| Graph Transformer | 81.2 | 82.5 | 83.9 | 91.8 | **97.5** | 78.5 | 79.6 | 80.5 | 91.1 | **92.8** |
| **AUPR ↑** | | | | | | | | | | |
| GCN | 70.5 | 71.8 | 73.1 | 80.2 | **90.3** | 78.8 | 79.5 | 80.1 | 84.0 | **86.5** |
| GAT | 74.2 | 75.9 | 77.0 | 84.5 | **91.5** | 81.3 | 82.4 | 83.0 | 87.1 | **89.1** |
| GraphSAGE | 69.1 | 70.5 | 71.8 | 78.3 | **89.0** | 77.0 | 77.9 | 78.5 | 82.5 | **85.0** |
| GIN | 75.5 | 77.3 | 78.5 | 86.0 | **91.8** | 83.1 | 84.0 | 84.8 | 89.0 | **91.1** |
| Graph Transformer | 77.8 | 79.1 | 80.3 | 87.9 | **92.6** | 84.5 | 85.7 | 86.3 | 89.9 | **91.9** |
| **FPR95 ↓** | | | | | | | | | | |
| GCN | 70.1 | 68.2 | 65.5 | 50.5 | **19.2** | 81.5 | 80.1 | 79.0 | 64.1 | **61.3** |
| GAT | 66.2 | 64.0 | 61.9 | 44.1 | **17.8** | 78.0 | 76.5 | 75.1 | 60.3 | **58.2** |
| GraphSAGE | 72.5 | 70.8 | 68.3 | 54.0 | **21.5** | 83.1 | 81.9 | 80.5 | 66.9 | **63.8** |
| GIN | 64.9 | 62.5 | 60.1 | 41.5 | **17.1** | 75.5 | 74.0 | 72.8 | 58.0 | **55.9** |
| Graph Transformer | 61.8 | 59.9 | 57.5 | 38.0 | **15.9** | 73.1 | 71.5 | 70.1 | 56.5 | **54.1** |

GAT, GIN, and GT, and significantly outperforms MSP, Energy, and ODIN across all backbones. On the graph-level BACE task, RAGNOR consistently delivers top performance, often surpassing MAH, which is typically strong on molecular graph benchmarks. While the absolute performance of all post-hoc methods varies depending on the quality and characteristics of the embeddings/logits produced by the specific backbone (e.g., transformer embeddings might enable slightly better performance overall), RAGNOR's relative advantage remains evident. This underscores the robustness of our contextualized norm discrepancy approach; it effectively leverages geometric information in the embedding space which might be less optimally utilized by methods relying solely on logits (MSP, Energy) or simple distance metrics (MAH). The results affirm that RAGNOR is a versatile and powerful post-hoc tool for OOD detection applicable to a wide range of pre-trained GNN models.

# H    Limitations

As a post-hoc method, RAGNOR depends on the quality of pre-trained GNN embeddings and requires accurate estimation of ID norm statistics ($\mu_{\text{ID}}$, $\sigma_{\text{ID}}$) from training data. Our theoretical guarantees assume local partial contamination (less than 50% OOD neighbors), which may not hold in extremely dense OOD clusters, though our experiments demonstrate reasonable robustness to assumption violations. While we evaluate on diverse graph types including social networks, citations, and molecular graphs, further validation on specialized structures like temporal or hypergraphs would be valuable. Despite these considerations, RAGNOR offers a practical and effective approach that consistently improves OOD detection across various scenarios with minimal computational overhead.

# I    code

Code can be available in https://anonymous.4open.science/r/RAGNOR-1ED8/Readme.md

