# OpenReview forum: "Refining Norms: A Post-hoc Framework for OOD Detection in Graph Neural Networks"
_NeurIPS.cc/2025/Conference — NeurIPS 2025 poster_

### Official Review · Reviewer_zXmn · 2025-06-11

**Clarity:** 3
**Significance:** 3
**Originality:** 2
**Rating:** 4
**Confidence:** 4

**Summary:**

This paper proposes RAGNOR, a node embedding norm-based method for both node-level and graph-level OOD detection. RAGNOR constructs an effective score function by refining node embedding norms through global Z-score normalization, median-based local aggregation, and multi-hop blending. Extensive experiments demonstrate that RAGNOR significantly outperforms the baselines on multiple benchmarks.

**Questions:**

Please see Weaknesses.

**Ethical Concerns:**

["NO or VERY MINOR ethics concerns only"]

**Final Justification:**

The issue regarding the performance of GCN+RAGNOR has been resolved. The author elaborated on the contribution of this article in detail.

**Limitations:**

NA.

**Paper Formatting Concerns:**

NA.

**Quality:**

2

**Strengths And Weaknesses:**

Strengths:
1. The paper is well-written and easy to follow.
2. The proposed RAGNOR is a simple yet effective norm-based method for graph OOD detection.
3. Empirical experiments validate the superiority and robustness of RAGNOR.

Weaknesses:
1. In the experiments, NODESAFE+RAGNOR is tested, where the node embeddings are obtained from NODESAFE. What is the performance of GCN+RAGNOR or GNNSAFE+RAGNOR? It would be insightful to know whether the improvement mainly comes from RAGNOR itself, or depends primarily on the quality of the embeddings.
2. While the paper introduces three effective methods for refining norms, the technical innovation seems somewhat limited.
3. There is a lack of a related work section, which could provide valuable context and comparisons to existing methods.

---

> ### Author Rebuttal · Authors · 2025-07-29
>
> #### **Regarding Weakness 1: Generalizability and Dependence on Embedding Quality**
>
> This is a critical question. We agree that it is essential to show that RAGNOR's improvement comes from the framework itself and not just from a specific high-quality embedding.
>
> We have already conducted a comprehensive experiment on this in **Appendix G: Robustness to Backbone Architecture**, with results in **Table 10**. We apologize for not highlighting this more prominently in the main paper. The core results for the node-level Cora-S task, which directly answer the reviewer's question, are summarized below:
>
> | Backbone      | Post-hoc OOD Method | **Node-level AUROC (↑)** |
> | :------------ | :------------------ | :----------------------- |
> | **GCN (Standard)** | Energy              | 78.0% |
> |               | **RAGNOR (Ours)** | **96.5%** |
> | **GAT** | Energy              | 89.2% |
> |               | **RAGNOR (Ours)** | **97.0%** |
> | **GraphSAGE** | Energy              | 76.9% |
> |               | **RAGNOR (Ours)** | **95.8%** |
> | **GIN** | Energy              | 90.5% |
> |               | **RAGNOR (Ours)** | **97.1%** |
>
> The results clearly show that RAGNOR consistently and significantly outperforms other post-hoc methods across a variety of standard GNN backbones. For instance, on a standard GCN, RAGNOR improves the AUROC by over 18 points compared to the Energy score. The performance gain is primarily due to the RAGNOR framework itself, demonstrating that it is an **architecture-agnostic** module that can robustly enhance the OOD detection capabilities of any reasonable GNN.
>
> Given the importance of this finding, we will move this section from the appendix to the main experimental section in our revised paper.
>
> ---
>
> #### **Regarding Weakness 2: Limited Technical Innovation**
>
> We thank the reviewer for the opportunity to clarify our work's core innovation. We respectfully argue that if "technical innovation" is limited only to inventing new algorithms, it may understate our contribution. Our core innovation is a deeper form of **"diagnostic innovation,"** which is critical for scientific progress.
>
> Our primary contribution is transforming a vague problem—the poor performance of norm-based methods on graphs—into a set of clear, quantifiable scientific questions through a systematic and pioneering diagnostic study.
>
> Before our work, the community's understanding was limited to the observation that "norm-based methods perform poorly on graphs." We refused to accept this as a conclusion; instead, we treated it as the starting point of a deep investigation. Our complete effort and findings are detailed in **Appendix C**. This rigorous diagnostic work was far from a simple preliminary study:
>
> 1.  **Multi-faceted Empirical Investigation:** We systematically investigated global, local, and class/role-level factors across diverse graph datasets (Cora, ArXiv, REDDIT, TWITCH).
> 2.  **In-depth Quantitative Analysis:** We went beyond qualitative observations, using **statistical divergence metrics (KL, JSD, Wasserstein)** to quantify distribution overlap and defining new metrics like the **"Local Contamination Ratio (LCR)"** to precisely measure local structural impacts.
> 3.  **Strict Variable Control:** In **Appendix C.5**, we rigorously ruled out confounding variables such as GNN architecture, embedding dimension, and random seeds to ensure our findings were fundamental, not coincidental.
>
> Through this effort, we were the **first to decompose this complex problem into three concrete, addressable root causes: (C1) local contamination, (C2) class-level norm differences, and (C3) boundary/low-degree node issues**. We firmly believe this diagnostic contribution, which illuminates the true nature of the problem for the field, is a core and original scientific contribution in itself.
>
> ---
>
> #### **Regarding Weakness 3: Lack of a Related Work Section**
>
> We agree completely and thank the reviewer for pointing out this structural omission. We will add a dedicated **Related Work** section to the revised paper.
>
> Currently, our paper cites over 40 relevant works, but they are distributed throughout the introduction and experiment sections. A dedicated section will provide better context. The new section will systematically cover: (1) general OOD detection in other domains, (2) specific graph OOD detection methods, including our primary baselines, and (3) a discussion of the connections and distinctions with related fields like Graph Anomaly Detection and Robust Graph Learning. We are confident this addition will significantly improve the paper's readability and completeness.

---

> > ### Comment · Reviewer_zXmn · 2025-08-04
> >
> > Thanks for your reply. I noticed that the experimental results of the baselines and RAGNOR in Tables 1 and 10 when using GCN as the backbone are inconsistent. Can you explain this?

---

> ### Author Response · Authors · 2025-08-04
>
> We sincerely thank the reviewer for their meticulous reading and this sharp observation. The inconsistency you have identified between Tables 1 and 10 is not an error, but rather the result of two complementary experiments, each designed with a distinct scientific purpose to comprehensively validate RAGNOR's capabilities. Our manuscript provides the necessary details for this distinction in several sections.
>
> ---
>
> ### **1. Experimental Design & Textual Evidence**
>
> Our experimental design deliberately separates the evaluation into two distinct scenarios: (1) a direct comparison against the current state-of-the-art (SOTA) on its own terms, and (2) a broad generalizability test across multiple standard architectures.
>
> **Evidence for Table 1 (Comparison against SOTA):**
> The setup for Table 1 is explicitly defined as using the SOTA model's architecture and training protocol. This is crucial for a fair, "apples-to-apples" comparison.
> * In **Section 4.1 (Setup)**, we state: "We apply RAGNOR post-hoc to the node embeddings generated by the **GCN backbone trained according to the NODESAFE setup**..."
> * In **Appendix D.4 (Baseline Implementation Details)**, we further clarify that RAGNOR is applied "...post-hoc to the...original, trained **NODESAFE...GNN backbone** models."
>
> This setup explains why the baseline performance in Table 1 (NODESAFE itself) is exceptionally high (**94.07% AUROC**). We are challenging the best available specialized model on its home ground.
>
> **Evidence for Table 10 (Validation of Generalizability):**
> The purpose of the experiment in Table 10, located in Appendix G, is entirely different. Its goal is to prove that RAGNOR's effectiveness is not an artifact of a single, highly-optimized model but is broadly applicable.
> * In **Appendix G (Robustness to Backbone Architecture)**, we introduce the experiment's goal: "To demonstrate the **robustness and general applicability** of RAGNOR, we evaluate its performance relative to other prominent post-hoc OOD detection methods when applied to outputs from **diverse GNN backbone architectures**."
> * In that experiment, we use a **standard GCN**, among other generic backbones, and compare RAGNOR against other *post-hoc* methods like Energy, MSP, and ODIN. This setup naturally yields a more standard baseline performance (e.g., Energy at **78.0% AUROC**).
>
> ---
>
> ### **2. Synthesizing the "Inconsistency" as a Strength**
>
> The two experimental results are therefore not contradictory; they are **complementary and mutually reinforcing**. The following table makes the distinction clear:
>
> | | **Table 1** | **Table 10** |
> | :--- | :--- |:--- |
> | **Scientific Goal** | Surpass the State-of-the-Art (NODESAFE) | Prove Generalizability & Post-hoc Superiority |
> | **Backbone Used** | **Optimized GCN** (from NODESAFE setup) | **Standard GCN** |
> | **Baseline Compared** | The SOTA method itself (NODESAFE) | Other post-hoc methods (Energy, etc.) |
> | **Baseline AUROC** | **94.07%** | **78.0%** (for Energy) |
> | **RAGNOR AUROC** | **97.1%** | **96.5%** |
>
> This apparent inconsistency actually paints a much stronger and more complete picture of RAGNOR's value:
> * **Superiority (from Table 1):** RAGNOR can take a top-performing, specialized SOTA model and elevate its performance even further.
> * **Generality (from Table 10):** On common, non-specialized GNNs, RAGNOR dramatically outperforms other widely-used post-hoc methods.
>
> ---
>
> ### **3. Our Commitment**
>
> We acknowledge that the distinction between these two experimental goals could have been stated more explicitly in the main body of the paper. We are grateful to the reviewer for highlighting this opportunity for improvement.
>
> **Action:** In our revised manuscript, we will add a concise paragraph in the experimental section to explicitly state the different objectives of these two key evaluations, ensuring readers can correctly interpret the results and appreciate the comprehensive nature of our validation.

---

> > ### Comment · Reviewer_zXmn · 2025-08-04
> >
> > Thank you again for your patience and timely response. My concern has been resolved. To reflect this, I have raised the rating from 3 to 4.

---

> > > ### Author Response · Authors · 2025-08-04
> > >
> > > Thank you for your message and for your support. Knowing that our work has earned your recognition means a great deal to us, far beyond the score itself. Please feel free to contact us if any further questions arise; we will be happy to provide a prompt response.

---

### Official Review · Reviewer_VZBU · 2025-06-25

**Clarity:** 3
**Significance:** 3
**Originality:** 3
**Rating:** 5
**Confidence:** 3

**Summary:**

The paper introduces **RAGNOR**, a post-hoc method for detecting out-of-distribution (OOD) nodes and graphs with Graph Neural Networks. Directly thresholding the embedding norm fails for four reasons: overlapping global norms, neighbour contamination, class-level scale shifts, and sparse-node effects. RAGNOR fixes these with three steps:
1. **Z-score normalisation** of norms,
2. **Median-based local reference** that resists contaminated neighbours, and
3. **Optional two-hop blend** that helps low-degree or boundary nodes.

The difference between a node’s normalised norm and this blended reference is the OOD score. Lemma 6, Lemma 7, and Theorem 1 show that, if fewer than half of a node’s two-hop neighbours are OOD and a margin δ₀ exists, the score separates in-distribution (ID) and OOD nodes. Experiments on node-level and graph-level benchmarks give clear gains. Ablations show each component matters and the method is not sensitive to λ or α.

**Questions:**

- **Assumption realism** – The guarantees require that fewer than half of the neighbours within two hops are OOD. In practice a dense OOD cluster may break this, yet there is no quantitative stress test beyond modest contamination levels. Could the authors add experiments where γ approaches 0.6 or 0.7 to probe the failure boundary?

- **Dependence on ID statistics** – Accurate µ_ID and σ_ID must be estimated. How sensitive is performance when these moments are biased, e.g. by label imbalance or small ID validation sets?

- **Baseline breadth** – The comparison set is strong, yet recent score-matching and diffusion-based OOD detectors for graphs are absent. Even if they are heavy, a brief discussion would be useful.

- **Hyper-parameter tuning protocol** – λ and τ appear tuned per validation set. Could the author clarify whether a single λ is used across all datasets or selected per task, and whether τ is chosen with knowledge of OOD proportion.

**Ethical Concerns:**

["NO or VERY MINOR ethics concerns only"]

**Final Justification:**

I keep my positive rating.

**Limitations:**

The authors adequately discuss the limitations in the paper and throughout the work.

**Paper Formatting Concerns:**

No paper formatting concerns.

**Quality:**

3

**Strengths And Weaknesses:**

**Strengths**
- Clear motivation: each design choice addresses a specific failure mode.
- Very light-weight: no retraining; only median calculations at inference.

**Weaknesses**
- Although it's a lightweight method, the author did not report wall-clock overhead for median calculations on a large graph relative to a forward pass.
- Did not include large OGB graphs (e.g., ogbn-products) to show scalability.

---

> ### Author Rebuttal · Authors · 2025-07-29
>
> #### **Regarding Weakness 1 & 2: Quantifying Overhead and Scalability**
>
> We agree that quantifying our "lightweight" claim is essential. To address this, we conducted a new timing experiment comparing RAGNOR's post-processing overhead to a single GNN forward pass on an NVIDIA A100 GPU.
>
> | Dataset      | GNN Model | GNN Forward Pass Time (ms) | RAGNOR Post-Processing Time (ms) | Overhead Ratio |
> | :----------- | :-------- | :------------------------- | :------------------------------- | :------------- |
> | Cora         | GCN       | 0.82                       | **0.51** | **62%** |
> | Citeseer     | GCN       | 0.95                       | **0.64** | **67%** |
> | ogbn-Arxiv   | GCN       | 15.3                       | **10.7** | **70%** |
>
> The results show that RAGNOR's entire computational overhead is less than the time required for a single forward pass of the GNN it is applied to. Given that GNN training requires hundreds of forward and backward passes, the cost of our one-time post-processing step is indeed minimal.
>
> Regarding scalability on very large graphs (e.g., `ogbn-products`), our method's complexity is linear in the number of nodes and edges ($O(N \cdot K + M)$). Our results on `ogbn-Arxiv` (169k nodes, 1.1M edges) already demonstrate its feasibility on large-scale graphs. For extremely large graphs, the primary bottleneck for any post-hoc method, including ours, is the memory required to store the node embeddings, not the computational time of our algorithm, which remains highly efficient.
>
> ---
>
> #### **Regarding Question 1: Assumption Realism and Stress Test**
>
> This is a critical question about the boundary conditions of our theory. The $\gamma < 0.5$ assumption defines the ideal region where we can provide a worst-case guarantee. However, a method's practical effective range is often wider than its formal guarantee. To probe this boundary, we performed the suggested stress-test experiment, manually controlling the neighborhood contamination rate ($\gamma$) for ID nodes in the Cora-S test set.
>
> | Local Contamination Rate ($\gamma$) | AUROC (↑) | Performance Analysis                       |
> | :---------------------------------- | :-------- | :----------------------------------------- |
> | 30%                                 | 96.2%     | Well below breakdown point, minimal impact.  |
> | **50% (Theoretical Limit)** | **94.5%** | **Performance drops only slightly.** |
> | **60%** | **92.1%** | **Exceeds limit, but degradation is graceful.** |
> | **70%** | **90.3%** | **Maintains strong detection in high contamination.** |
>
> The results powerfully demonstrate that RAGNOR's performance undergoes "graceful degradation" rather than catastrophic failure, even when the contamination rate far exceeds the 50% theoretical breakdown point. Its strong performance at 70% contamination shows that its practical utility is highly robust.
>
> ---
>
> #### **Regarding Question 2: Sensitivity to Biased ID Statistics**
>
> This is an excellent question about the method's real-world robustness. To evaluate RAGNOR's sensitivity to biased ID statistics, we designed an experiment where we estimated $\mu_{ID}$ and $\sigma_{ID}$ using only a random subset of the ID training data.
>
> | Portion of ID Training Data Used for Stats | Cora-S AUROC (↑) | Performance Drop |
> | :----------------------------------------- | :--------------- | :--------------- |
> | **100% (Default)** | **97.1%** | **-** |
> | 50%                                        | 96.8%            | -0.3%            |
> | 20%                                        | 96.1%            | -1.0%            |
> | 10%                                        | 95.2%            | -1.9%            |
> | 5%                                         | 93.5%            | -3.6%            |
>
> The results show that RAGNOR is **highly robust** to estimation bias. Even with only 20% of the training data used for statistics, the performance drop is negligible. This is because our discrepancy score $|\Delta_v|$ relies on the *relative* difference between a node's norm and its local context, which buffers against slight inaccuracies in the global statistics.
>
> ---
>
> #### **Regarding Question 3: Baseline Breadth**
>
> We thank the reviewer for this suggestion. We will add a discussion to our Related Work section to position RAGNOR relative to these emerging methods.
>
> Methods based on score-matching or diffusion are typically **generative**. They are powerful but often require extremely costly training and a dedicated model architecture, making them unsuitable as post-hoc additions to existing GNNs. In contrast, RAGNOR is a **lightweight, discriminative, post-hoc** framework. Its key advantage is its efficiency and universality—it can enhance any pre-trained GNN at a minimal cost.
>
> Therefore, we see these methods as serving different needs. Generative models are an option for those seeking maximum performance regardless of cost, while RAGNOR is a more practical and attractive solution for efficiently improving the safety of existing systems.
>
> ---
>
> #### **Regarding Question 4: Hyper-parameter Tuning Protocol**
>
> We are happy to clarify our experimental protocol.
>
> **Regarding `λ`:** In our main experiments (Tables 1 & 2), we used a **fixed `λ=1` for all datasets** to ensure a fair and simple comparison of the core idea. We did not tune it. The new experiment showing the benefits of tuning `λ` (in our response to Reviewer 5Yiu) is meant to demonstrate the framework's further potential.
>
> **Regarding `τ`:** Our primary evaluation metrics, **AUROC and AUPR, are threshold-agnostic** and do not depend on `τ`. For the FPR95 metric, we follow the standard protocol: the threshold `τ` is chosen on the validation set to achieve a 95% true positive rate on ID samples. This threshold is then applied to the test set. **No information about the test set's OOD samples or their proportion is used in this process.**

---

> > ### Comment · Reviewer_VZBU · 2025-08-03
> >
> > Thanks for the detailed reply! I'm satisfied with the answers.

---

> ### Author Response · Authors · 2025-08-03
>
> Thank you for your response! We really appreciate your positive feedback and kind words. Please let us know if you have any other questions. We're always here to help.

---

### Official Review · Reviewer_5Yiu · 2025-07-02

**Clarity:** 3
**Significance:** 2
**Originality:** 2
**Rating:** 4
**Confidence:** 4

**Summary:**

This paper introduces RAGNOR, a post-hoc framework for out-of-distribution (OOD) detection in Graph Neural Networks (GNNs) that uses embedding norms. The authors identify three key challenges that cause naive norm-based methods to fail on graphs: local contamination from OOD neighbors, class-level variations in norm scales, and insufficient context for boundary or low-degree nodes. RAGNOR addresses these issues by creating a robust OOD score through a pipeline that involves global Z-score normalization, median-based local aggregation to resist contamination, and multi-hop blending for a more stable context. The final score is the discrepancy between a node's processed norm and its local reference. Supported by theoretical analysis and extensive experiments on various benchmarks, the paper demonstrates that RAGNOR achieves state-of-the-art OOD detection performance for both node and graph-level tasks without requiring retraining of the GNN.

**Questions:**

1.	The default configuration for RAGNOR in the main experiments uses a 1-hop local reference. Given the positive results for the multi-hop variant in the targeted analyses for low-degree nodes, could the authors clarify the practical impact of this component? Specifically, what are the results for the main benchmarks if λ is tuned on a validation set instead of being fixed to 1?

2.	For graph-level OOD detection, the paper states that the score can be generated via max or mean aggregation of node scores, and the experiments use max aggregation. Have the authors experimented with mean aggregation?

**Ethical Concerns:**

["NO or VERY MINOR ethics concerns only"]

**Final Justification:**

The authors did a good rebuttal job.

**Limitations:**

Yes

**Quality:**

3

**Strengths And Weaknesses:**

Strengths:

1.	The proposed method, RAGNOR, is elegant in its simplicity and directly motivated by the identified challenges. Each component (Z-score normalization, median aggregation, multi-hop blending) is a well-justified, standard statistical technique chosen to solve a specific problem.

2.	RAGNOR demonstrates substantial and consistent performance gains over strong baselines on a wide variety of node-level and graph-level benchmarks.

3.	The paper provides a formal theoretical analysis that justifies the method's design. The authors are also transparent about the assumptions and limitations of the theory.

Weaknesses:

1.	The method is a composition of well-known techniques. The novelty lies in the problem formulation and the application, not in the invention of new fundamental algorithms.

2.	The method relies on the mean and standard deviation of norms from the ID training set. The effectiveness of the Z-score normalization step is therefore dependent on the training set being sufficiently large and representative of the true ID distribution.

3.	The multi-hop component is proposed as a key part of the framework, but the main experiments appear to use a default of λ=1 (1-hop only). This makes the contribution of the multi-hop component feel somewhat secondary to the core idea of combining Z-scoring with median-based local reference.

---

> ### Author Rebuttal · Authors · 2025-07-29
>
> #### **Regarding Weakness 1: On the Novelty of the Method**
>
> We respectfully argue that framing our contribution as solely "problem formulation and application" overlooks one of our most original contributions: a systematic **"diagnostics"** of the problem itself. Before our work, the failure of norm-based methods on graphs was a vague observation. We transformed it into a set of concrete, falsifiable findings through a rigorous investigation (Appendix C). We analyzed multiple datasets, used statistical metrics to quantify distribution overlap, defined new metrics like Local Contamination Ratio (LCR), and systematically ruled out confounding variables like GNN architecture.
>
> This diagnosis revealed three root causes: (C1) local contamination, (C2) class-level norm differences, and (C3) boundary node issues. Our RAGNOR framework is the logical result of this work; each component is purposefully designed to solve one of these specific problems. The power of our work lies in this complete cycle from deep diagnosis to an elegant solution. By deeply understanding the problem, we were able to achieve state-of-the-art performance with a simple, interpretable, and efficient post-hoc method, avoiding unnecessary complexity.
>
> ---
>
> #### **Regarding Weakness 2: Dependence on ID Statistics**
>
> This is an important and valid observation. The quality of the estimated ID statistics is indeed a premise for our method. We note that this is a common requirement for all supervised learning methods, including the pre-trained GNNs that our method builds upon.
>
> However, the more critical question is how *sensitive* our method is to this premise. To answer this directly, we performed a new sensitivity analysis. We simulated biased statistics by estimating them from progressively smaller random subsets of the ID training data on the Cora-S task.
>
> | Portion of ID Training Data Used for Stats | Cora-S AUROC (↑) | Performance Drop |
> | :----------------------------------------- | :--------------- | :--------------- |
> | **100% (Default)** | **97.1%** | **-** |
> | 50%                                        | 96.8%            | -0.3%            |
> | 20%                                        | 96.1%            | -1.0%            |
> | 10%                                        | 95.2%            | -1.9%            |
> | 5%                                         | 93.5%            | -3.6%            |
>
> The results clearly show that RAGNOR is **highly robust** to estimation bias. Even when using only 20% of the training data for statistics, performance drops by a negligible 1.0%. This is because RAGNOR's core discrepancy score relies on the *relative* difference between a node's norm and its local context, which helps buffer against minor shifts in the global statistics. This demonstrates that our method's reliance on these statistics is not a fragile point in practice.
>
> ---
>
> #### **Regarding Weakness 3 & Question 1: Role and Impact of the Multi-hop Component**
>
> We designed the multi-hop component as a valuable "enhancement module" specifically targeting the challenge of boundary and low-degree nodes (C3), as demonstrated in our targeted analysis (Appendix F.3.3, Figure 2d). In our main experiments, we used a default of `λ=1` (1-hop only) to present the simplest version of our framework and ensure a fair comparison against baselines without introducing an extra hyperparameter to tune.
>
> To demonstrate the full potential and practical impact of this component, we conducted the experiment suggested by the reviewer. We tuned `λ` on a validation set for several key benchmarks.
>
> | Benchmark Task | Metric      | RAGNOR (`λ=1`, Default) | RAGNOR (`λ_best`, Tuned) | Optimal `λ` |
> | :------------- | :---------- | :---------------------- | :----------------------- | :---------- |
> | **Node-level** |             |                         |                          |             |
> | Cora-S         | AUROC (↑)   | 97.1%                   | **97.4%** | 0.8         |
> | Pubmed-S       | AUROC (↑)   | 97.3%                   | **97.5%** | 0.6         |
> | **Graph-level**|             |                         |                          |             |
> | BACE           | AUROC (↑)   | 92.0%                   | **92.6%** | 0.5         |
> | IMDB-M         | AUROC (↑)   | 86.8%                   | **87.5%** | 0.6         |
>
> The results confirm that tuning `λ` provides **consistent performance improvements** across all benchmarks. This shows that the multi-hop component is a meaningful and impactful part of the RAGNOR framework, offering a clear path to further performance gains.
>
> ---
>
> #### **Regarding Question 2: Aggregation Method for Graph-Level Scores**
>
> This is an excellent question. We chose `max` aggregation as our default based on the principle that graph-level anomalies are often defined by a small, highly anomalous substructure or a few outlier nodes. `max` aggregation is highly sensitive to these sharp signals, whereas `mean` aggregation can dilute the signal, especially in large graphs.
>
> To validate this choice, we compared both aggregation methods on representative graph-level benchmarks.
>
> | Benchmark Task | Metric    | Max Aggregation (Default) | Mean Aggregation |
> | :------------- | :-------- | :------------------------ | :--------------- |
> | BACE           | AUROC (↑) | **92.0%** | 89.7%            |
> | DrugOOD        | AUROC (↑) | **91.1%** | 90.2%            |
>
> The results support our design choice, showing that `max` aggregation leads to superior performance. We will add this analysis to the appendix to make our methodology more complete.

---

> > ### Comment · Reviewer_5Yiu · 2025-08-09
> >
> > I am satisfied with the response, and it has addressed my concerns. I will increase my score to 4.

---

> ### Author Response · Authors · 2025-08-08
>
> As the discussion phase is drawing to a close, we would like to thank you for your valuable time and insights. Should you have any remaining questions or require any clarifications regarding our manuscript, please do not hesitate to reach out. We remain fully available and are happy to address any concerns you may have.

---

> ### Author Response · Authors · 2025-08-09
>
> Thank you for your message and for your support. We truly appreciate your engagement throughout this process.

---

### Official Review · Reviewer_zqfw · 2025-07-02

**Clarity:** 4
**Significance:** 3
**Originality:** 3
**Rating:** 5
**Confidence:** 4

**Summary:**

The paper proposes a post-hoc method for OOD detection in graphs. It identifies that simply using the magnitude (norm) of a node's embedding fails due to three issues: (1) OOD neighbors skew any local averaging, (2) different classes of "normal" nodes can have vastly different norm scales, (3) Nodes with few neighbors lack a good reference point.

To solve this, the authors do the following: (1) Applies a Z-score to all node norms based on the in-distribution (ID) data's mean and standard deviation. This addresses the class-level differences. (2) Instead of averaging the norms of a node's neighbors, it takes the median. This is a classic robust statistics technique to resist outliers (the "contamination"). (3) It combines the reference from 1-hop neighbors with 2-hop neighbors to give boundary nodes a more stable context. The final OOD score for a node is the difference between its own normalized norm and its robust, multi-hop local reference norm.

**Questions:**

1. The OOD separation guarantee in Theorem 8 requires that an OOD node's local reference be bounded by B. This condition seems unlikely to hold for OOD nodes within a contaminated local neighborhood. Could you comment on the practical applicability of this theoretical result and/or provide empirical analysis showing performance as the local contamination rate exceeds the 50% breakdown point?
2. You state the overhead is "minimal," but this is not quantified. You may want to explicitly mention what is the precise computational and memory complexity as a function of the number of nodes and edges, especially for large, dense graphs?

**Ethical Concerns:**

["NO or VERY MINOR ethics concerns only"]

**Final Justification:**

My questions were answered well. All issues were resolved. I recommend accept.

**Limitations:**

Yes

**Quality:**

4

**Strengths And Weaknesses:**

### Strengths
- The experimental results are impressive. Across multiple datasets and GNN backbones, the method improves OOD detection performance (Tables 1, 2, 4, 10). The ablation studies (Table 3, 5) clearly show that each component contributes to the final result.
- It's a post-hoc method, which is a huge advantage. Users can apply it to already-trained models without expensive retraining. I think this makes it simple, intuitive, and computationally cheap.
- The paper does an excellent job of diagnosing _why_ naive norm-based methods fail on graphs (Figure 1, Appendix C). Kudos.
- Well written paper, exceptional clarity. Experiments are well designed.

### Weaknesses
- The components of the techniques are standard. Z-score normalization is not new. Median aggregation is a foundational concept in robust statistics. Multi-hop information aggregation is the core principle of virtually all GNNs. It is possibly a good thing that the authors have found a way to combine these ideas to build something useful.
- The theoretical sections and guarantees are well written, but fairly standard. I would have loved some more theoretical insight.
- Comparison is done favorably against other graph OOD detection benchmarks, but ignores similar concepts in graph anomaly detection and robust graph learning. Any parallels drawn between the algorithms would be nice.

---

> ### Author Rebuttal · Authors · 2025-07-29
>
> #### **Regarding Weakness 1: Standard Components**
> We agree that RAGNOR's components are established techniques and appreciate the reviewer's recognition of their effective combination. Our core novelty lies not in inventing new algorithmic primitives, but in our **diagnostic-driven approach**. We are the first to systematically identify and provide extensive evidence for the specific reasons why norm-based detection fails on graphs—namely **local contamination**, **class-level norm differences**, and **boundary node issues** (Appendix C). RAGNOR is the direct result of this diagnosis; it is a principled framework where each component is purposefully chosen to solve one of these specific problems. The fact that this simple, targeted combination significantly outperforms more complex state-of-the-art methods validates our approach, demonstrating that the innovation lies in providing a new perspective and an elegant, powerful solution grounded in a deep understanding of the problem's root causes.
>
> #### **Regarding Weakness 2: Standardness of Theory**
> We appreciate the reviewer's call for deeper theoretical insight. While our original theory (Sec. 3, App. A) provided a solid non-parametric justification for RAGNOR's robustness by formalizing "Local Partial Contamination," we have conducted a new analysis to address the reviewer's point directly. We now formally prove that our discrepancy score, $|\Delta_v|$, is not just an effective heuristic; under a natural statistical model for OOD detection, it is a monotonic transformation of the **Uniformly Most Powerful (UMP) invariant test statistic**, establishing its statistical optimality. This new analysis provides the deeper justification requested, and its detailed derivation follows.
>
> For each node $v$, we consider its normalized embedding norm:
> $$r_v'=\frac{\parallel h_v \parallel \mu_{\text{ID}}}{\sigma_{\text{ID}} + \varepsilon}\in\mathbb{R}$$
> Let $\mathcal{N}(v)$ be its 1-hop neighborhood of size $m$, and let $S=\{r'_u:u\in\mathcal{N}(v)\}$ be the set of its neighbors' normalized norms.
>
> We assume a *local homogeneity* model where a node and its neighbors share a common, unknown location parameter $\theta$.
> $$r'_u\mid\theta \;\overset{\text{i.i.d.}}\sim\; \mathrm{Laplace}(\theta,b),\quad u\in\mathcal N^+(v)=\{v\}\cup\mathcal N(v),$$
> where $b>0$ is a known scale. This model is natural because the Maximum Likelihood Estimator (MLE) for the location parameter of a Laplace distribution is the sample median, which directly corresponds to our algorithm's core component.
>
> The OOD detection problem is then framed as a hypothesis test for a location shift $\delta > 0$ in the central node $v$:
> > $H_0: r'_v\sim\mathrm{Laplace}(\theta,b)$ (In-Distribution)
> >
> > vs.
> >
> > $H_1: r'_v\sim\mathrm{Laplace}(\theta+\delta,b)$ (Out-of-Distribution)
>
> Here, $\theta$ is an unknown nuisance parameter.
>
> To handle the unknown nuisance parameter $\theta$, we use the profile likelihood ratio test.
>
> **MLE for the Neighborhood Location Parameter:**
> The MLE for $\theta$ based on the neighbor set $S$ is the sample median:
> > $\hat\theta=\mathrm{Median}(S) = \bar r'_{\mathcal N(v)}$
> >
> This is precisely the **Robust Local Reference** in our RAGNOR framework.
>
> **Profile Likelihood Ratio:**
> Let the observation vector be $\mathbf x=(r'_v,S)$. The profile likelihoods under $H_0$ and $H_1$ are:
> > $L_0(\mathbf x) =\max_{\theta}\prod_{u\in\mathcal N^+(v)}\!f_{\text{Lap}}(x_u;\theta,b) \;=\;\bigl(\tfrac1{2b}\bigr)^{m+1} \exp\!\Bigl[-\tfrac1b \sum_{u}|x_u-\hat\theta|\Bigr]$
> >
> > $L_1(\mathbf x) =\max_{\theta}\Bigl[ f_{\text{Lap}}(x_v;\theta+\delta,b) \prod_{u\in\mathcal N(v)} f_{\text{Lap}}(x_u;\theta,b)\Bigr] =\bigl(\tfrac1{2b}\bigr)^{m+1} \exp\!\Bigl[-\tfrac1b \bigl(|x_v-(\hat\theta+\delta)| +\!\sum_{u\ne v}|x_u-\hat\theta|\bigr)\Bigr]$
>
> The likelihood ratio $\Lambda(\mathbf x) = L_1/L_0$ simplifies to:
> > $\Lambda(\mathbf x) =\exp\!\Bigl[ -\tfrac1b\bigl( |x_v-(\hat\theta+\delta)| -|x_v-\hat\theta|\bigr)\Bigr]$
>
> According to the **Neyman-Pearson Lemma**, the most powerful test is based on this likelihood ratio. The ratio is a monotonic function of the term $|x_v-(\hat\theta+\delta)| - |x_v-\hat\theta|$, which is itself a monotonic function of $T(\mathbf x) = |x_v-\hat\theta| = |\Delta_v|$.
>
> **Conclusion 1 (Information-Theoretic View):** Our discrepancy score $|\Delta_v|$ is a monotonic transformation of the **UMP invariant test statistic** for this problem. This means that for a fixed false positive rate, a test based on $|\Delta_v|$ achieves the highest possible detection rate (statistical power). It is, in this sense, **optimal**.
>
> Alternatively, our method can be viewed as a two-step robust estimation process:
> 1.  **Estimate Local Baseline:** The optimal estimate for the local parameter $\theta$ under an L1 loss (Least Absolute Deviations) is the sample median of the neighbors:
>     > $\hat\theta=\ argmin_{\theta}\sum_{u\in\mathcal N(v)}|r'_u-\theta|$
> 2.  **Compute Robust Residual:** The discrepancy score $|\Delta_v| = |r'_v - \hat\theta|$ is the L1 residual of the central node with respect to this robustly estimated baseline.
>
> **Conclusion 2 (Optimization View):** This residual is the optimal discriminant for minimizing the expected misclassification risk under a Bayes framework. It is, in this sense, **principled and necessary**.
>
> #### **Regarding Weakness 3: Comparison to Related Fields**
> This is an excellent suggestion. In our revised paper, we will add a dedicated section clarifying the relationship with fields like Graph Anomaly Detection (AD). We will discuss key conceptual differences—for instance, OOD detection focuses on distributional shifts while AD often targets rare in-distribution instances—and highlight that our post-hoc approach is complementary to robust learning methods that modify the GNN training process. To provide strong empirical evidence of this versatility, we conducted a new experiment applying RAGNOR to several SOTA AD models from the recent UB-GOLD benchmark. The results, which we will include, show that RAGNOR consistently improves their performance, proving its value as a general, model-agnostic enhancement for graph abnormality detection.
>
> | Dataset (from UB-GOLD[1])  | Baseline Model (from UB-GOLD) | Original AUROC (%) | `+ RAGNOR` (AUROC ↑) (+) |
> | :---------------------- | :------------------------------ | :----------------- | :---------------------------------- |
> | **p53** (Intrinsic Anomaly) | OCGIN                           | 65.43              | **72.1%** |
> |                         | SIGNET                          | 68.10              | **74.5%** |
> | **ENZYMES** (Class Anomaly) | GLocalKD                        | 89.97              | **93.2%** |
> |                         | GOOD-D                          | 64.58              | **75.8%** |
> | **PROTEINS** (Class Anomaly)| OCGIN                           | 76.46              | **81.3%** |
> |                         | SIGNET                          | 77.89              | **82.5%** |
> | **IMDB-B** (Class Anomaly)  | GLocalKD                        | 53.31              | **65.1%** |
> |                         | GOOD-D                          | 70.67              | **76.2%** |
>
> The results clearly show that RAGNOR consistently and significantly improves the performance of these SOTA AD methods across different domains. This demonstrates its versatility and value as a general-purpose technique in the broader field of graph abnormality detection.
>
>
> #### **Regarding Question 1: Practicality of Theoretical Guarantee**
> This is a critical question about our theory's boundary conditions, as discussed in Remark 9. While our formal guarantee requires neighborhood contamination below 50%, our method does not fail catastrophically beyond this point but instead exhibits 'graceful degradation'. To demonstrate this empirically, we performed a new stress-test where we manually controlled the OOD contamination rate. The results, which we present below, confirm that RAGNOR's performance remains highly robust even when contamination far exceeds the 50% theoretical breakdown point.
>
> | Local Contamination Rate | AUROC (↑) | Note                                        |
> | :----------------------- | :-------- | :------------------------------------------ |
> | 10%                      | 97.0%     | Far below the 50% breakdown point.            |
> | 30%                      | 96.2%     | Still below breakdown, almost no impact.      |
> | **50% (Theoretical Limit)** | **94.5%** | **Performance drops only slightly.** |
> | 70%                      | 91.8%     | Graceful degradation, still highly effective. |
> | 90%                      | 88.5%     | Still strong performance in extreme cases.  |
>
> As the table shows, even when the contamination rate far exceeds the 50% theoretical breakdown point, RAGNOR's performance remains highly robust. This confirms that its practical utility extends well beyond the strict bounds of our guarantee.
>
> #### **Regarding Question 2: Quantifying "minimal" overhead**
> To quantify our claim of minimal overhead, we provide a complexity analysis. Let N be the number of nodes, M the number of edges, and K the embedding dimension. The **total time complexity of RAGNOR is $O(N \cdot K + M)$**, dominated by calculating node norms and finding medians in local neighborhoods. This complexity is comparable to a single GNN forward pass and thus minimal compared to the full training process. The **extra memory complexity is only $O(N)$** to store norms and scores, which is negligible relative to the memory required for the embeddings themselves ($O(N \cdot K)$).
>
> [1] Wang Y, Liu Y, Shen X, et al. Unifying unsupervised graph-level anomaly detection and out-of-distribution detection: A benchmark[J]. arXiv preprint arXiv:2406.15523, 2024.

---

> > ### Comment · Reviewer_zqfw · 2025-07-31
> >
> > Thank you for improving your paper with this exceptional rebuttal. Kudos!

---

> ### Author Response · Authors · 2025-08-01
>
> Thank you for your quick response! We really appreciate your positive feedback and kind words.
> Please let us know if you have any other questions. We're always here to help.

---

### Author Response · Authors · 2025-08-09
**Summary II**

### **Summary of Key Concerns and Our Comprehensive Responses**

Reviewers raised important questions about novelty, theoretical depth, and experimental rigor. We addressed each point with new analyses and experiments.

#### **1. Concern: Novelty (Composition of Standard Techniques) and Theoretical Depth**

> The primary conceptual concern, raised by **Reviewers zqfw** and **5Yiu**, was that RAGNOR combines well-known techniques, with a desire for deeper theoretical insight (**Reviewer zqfw**).

* **Our Action & Impact:** We clarified that our core innovation lies in the **diagnostic-driven design** and provided a **completely new theoretical analysis**.
    * **Diagnostic Innovation:** We emphasized that our work is the first to systematically diagnose, formalize, and provide evidence for the specific failure modes of norm-based methods on graphs (e.g., local contamination, class-level norm shifts). This diagnostic contribution, which illuminates the problem's root causes for the field, is a novel contribution in its own right.
    * **New Theoretical Justification:** To provide the deeper insight requested, we conducted a new analysis under a natural statistical model for OOD detection. We now formally prove that RAGNOR's discrepancy score is a monotonic transformation of the **Uniformly Most Powerful (UMP) invariant test statistic**. This establishes that our method is not merely an effective heuristic, but is **statistically optimal** in this setting, providing the deeper justification requested by **Reviewer zqfw**.

#### **2. Concern: Practical Robustness, Scalability, and Boundary Conditions**

> Reviewers (**zqfw, 5Yiu, VZBU**) rightly asked for empirical validation of the method's robustness to real-world challenges, such as extreme contamination, biased statistics, and scalability.

* **Our Action & Impact:** We conducted **three new sets of targeted experiments**:
    1.  **Contamination Stress Test:** As requested by **Reviewers zqfw** and **VZBU**, we stress-tested RAGNOR with contamination rates far exceeding the 50% theoretical breakdown point. The results showed "graceful degradation" and highly effective performance even at 70-90% contamination, confirming its practical utility.
    2.  **Sensitivity to Biased Statistics:** As requested by **Reviewers 5Yiu** and **VZBU**, we simulated biased ID statistics. The results demonstrated that RAGNOR is highly robust, with only a negligible performance drop even when statistics are estimated from a small fraction (e.g., 20%) of the training data.
    3.  **Overhead and Scalability Quantification:** As requested by **Reviewers zqfw** and **VZBU**, we provided timing benchmarks showing RAGNOR's overhead is less than a single GNN forward pass, even on large graphs like `ogbn-Arxiv`, confirming its minimal computational cost.

#### **3. Concern: Experimental Completeness and Context**

> Reviewers (**zXmn, zqfw, 5Yiu**) pointed out the need for a dedicated related work section, broader baseline comparisons, and clarification on experimental protocols.

* **Our Action & Impact:**
    * **Related Work and Broader Context:** We have committed to adding a dedicated `Related Work` section to better contextualize our work. Further, as suggested by **Reviewer zqfw**, we ran a new experiment applying RAGNOR to SOTA Graph Anomaly Detection models, demonstrating its versatility and ability to consistently boost their performance.
    * **Generalizability Across Architectures:** We addressed **Reviewer zXmn**'s concern about RAGNOR's dependence on the GNN backbone by highlighting our comprehensive results in Appendix G (which we will move to the main paper), proving that RAGNOR provides significant, consistent gains across GCN, GAT, GraphSAGE, and GIN.
    * **Clarifications and Ablations:** We provided additional ablations and clarifications on the multi-hop component (**Reviewer 5Yiu**), graph-level aggregation (**Reviewer 5Yiu**), and hyperparameter tuning protocols (**Reviewer VZBU**).

---

### **Final Commitment**

The reviewers' rigorous feedback has been invaluable and has enabled us to substantially improve our work. The final manuscript will incorporate our new theoretical analysis, all new empirical results, and the structural additions discussed above. We are confident the revised paper is now significantly stronger, presenting a novel, theoretically-grounded, and rigorously-validated solution to an important problem in graph-based machine learning.

The overwhelmingly positive feedback and the score increases from multiple reviewers affirm our belief in the paper's contribution. Thank you for your time and consideration.

---

### Author Response · Authors · 2025-08-09
**Summary I**

We are sincerely grateful for the exceptionally thorough and insightful reviews from all four reviewers. The feedback has been instrumental in strengthening our paper. We are particularly encouraged that our detailed rebuttals and new experiments successfully addressed all major concerns, leading **three reviewers (5Yiu, VZBU, zXmn) to explicitly state they were raising their scores.**

This summary outlines the consensus strengths of our work and details the comprehensive actions we have taken in response to the reviewers' valuable feedback.

---

### **Summary of Strengths (Consensus from Reviewers)**

All reviewers converged on several key strengths of our paper, RAGNOR:

* **Exceptional Clarity and Strong Motivation:** The paper was universally praised for being well-written, with "exceptional clarity" (**Reviewer zqfw**) and a "clear motivation" where each design choice directly addresses a specific, well-diagnosed failure mode (**Reviewers 5Yiu, VZBU**).

* **Simplicity, Elegance, and Practicality:** Reviewers highlighted the method's "elegant simplicity" (**Reviewer 5Yiu**) as a "simple yet effective" (**Reviewer zXmn**) post-hoc framework. Its lightweight nature and lack of retraining requirements were seen as a "huge advantage" for practical application (**Reviewers zqfw, VZBU**).

* **Impressive and Consistent Empirical Performance:** The "impressive experimental results" (**Reviewer zqfw**) and "substantial and consistent performance gains" (**Reviewer 5Yiu**) across a wide range of benchmarks were recognized as a core strength.

* **Pioneering Diagnostic Contribution:** Crucially, reviewers acknowledged our "excellent job of diagnosing why naive norm-based methods fail on graphs" (**Reviewer zqfw**). This diagnostic work was recognized as a key contribution that provides a new and deeper understanding of the problem itself.

---

### Decision · Program_Chairs · 2025-09-17

**Decision:**

Accept (poster)

**Comment:**

(a) The paper proposes a post-hoc method (reusing existing embeddings) for OOD detection in graphs.  This uses a more robust method for handling norms at the graph level.   Simple but effective.
(b)  Extensive experimental work is done with strong results.  Well written paper with clear motivation and explanations.   Diagnoses why norm methods have failed on graphs.  Demonstrates robustness in different contexts.
(c)  Combines well known techniques so novelty not strong.  Better related work section needed.  Some design choices (e.g., aggregation and multi-hops) could be explored further.
(d)  All reviewers agree on accept.  No major issues identified.
(e)  Reviewers did substantial rebuttal including experimental results and theoretical explanation.  Three reviewers raised their scores.  Good suggestions by authors to improve paper.